# An Invitation to Deep Reinforcement Learning

## Abstract

Training a deep neural network to maximize a target objective has become the standard recipe for successful machine learning over the last decade. These networks can be optimized with supervised learning, if the target objective is differentiable. For many interesting problems, this is however not the case. Common objectives like intersection over union (IoU), bilingual evaluation understudy (BLEU) score or rewards cannot be optimized with supervised learning. A common workaround is to define differentiable surrogate losses, leading to suboptimal solutions with respect to the actual objective. Reinforcement learning (RL) has emerged as a promising alternative for optimizing deep neural networks to maximize non-differentiable objectives in recent years. Examples include aligning large language models via human feedback, code generation, object detection or control problems. This makes RL techniques relevant to the larger machine learning audience. The subject is, however, time intensive to approach due to the large range of methods, as well as the often very theoretical presentation. In this introduction, we take an alternative approach, different from classic reinforcement learning textbooks. Rather than focusing on tabular problems, we introduce reinforcement learning as a generalization of supervised learning, which we first apply to non-differentiable objectives and later to temporal problems. Assuming only basic knowledge of supervised learning, the reader will be able to understand state-of-the-art deep RL algorithms like proximal policy optimization (PPO) after reading this tutorial.

## 1 Introduction

The field of reinforcement learning (RL) is traditionally viewed as the art of learning by trial and error (Sutton & Barto, 2018). Reinforcement learning methods were historically developed to solve sequential decision making tasks. The core idea is to deploy an untrained model in an *environment*. This model is called the *policy* and maps inputs to actions. The policy is then improved by randomly attempting different actions and observing an associated feedback signal, called the *reward*. Reinforcement learning techniques have demonstrated remarkable success when applied to popular games. For example, RL produced world-class policies in the games of Go (Silver et al., 2016; 2018; Schrittwieser et al., 2020), Chess (Silver et al., 2018; Schrittwieser et al., 2020), Shogi (Silver et al., 2018; Schrittwieser et al., 2020), Starcraft (Vinyals et al., 2019), Stratego (Perolat et al., 2022) and achieved above human level policies in all Atari games (Badia et al., 2020; Ecoffet et al., 2021; Kapturowski et al., 2023) as well as Poker (Moravčík et al., 2017; Brown & Sandholm, 2018; 2019). While these techniques work well for games and simulations, their application to practical real-world problems has proven to be more difficult (Dulac-Arnold et al., 2020). This has changed in recent years, where a number of breakthroughs have been achieved by transferring RL policies trained in simulation to the real world (Bellemare et al., 2020; Degrave et al., 2022; Kaufmann et al., 2023) or by successfully applying RL to problems that were traditionally considered supervised problems (Ouyang et al., 2022; Fawzi et al., 2022; Mankowitz et al., 2023). It has long been known that any supervised learning (SL) problem can be reformulated as an RL problem (Barto & Dieterich, 2004; Jiang et al., 2023) by defining rewards that match the loss function. This idea has not been used much in practice because the advantage of RL has been unclear, and RL problems have been considered to be harder to solve. A key advantage of reinforcement learning over supervised learning is that the optimization objective does not need to be differentiable. To see why this property is important, consider the task of text prediction, at which models like ChatGPT had a lot of success recently. The large language models used in this task are pre-trained

using self-supervision (Brown et al., 2020) on a large corpus of internet text, which allows them to generate realistic and linguistically flawless responses to text prompts. However, self-supervised models like GPT-3 cannot directly be deployed in products because they are not optimized to predict helpful, honest, and harmless answers (Bai et al., 2022). So far, the most successful technique to address this problem is called reinforcement learning from human feedback (RLHF) (Christiano et al., 2017; Stiennon et al., 2020; Bai et al., 2022; Ouyang et al., 2022) in which human annotators rank outputs of the model and the task is to maximize this ranking. The mapping between the models outputs and a human ranking is not differentiable, hence supervised learning cannot optimize this objective, whereas reinforcement learning techniques can. Recently, RL was also able to claim success in code generation (Mankowitz et al., 2023) by maximizing execution speed of predicted code, discovering new optimization techniques. Execution speed of code can easily be measured, but not computed in a differentiable way. Derivative-free optimization methods (Hansen, 2016; Frazier, 2018) can also optimize non-differentiable objectives but typically do not scale well to deep neural networks. A second advantage RL has over supervised learning is that algorithms can collect their own data which allows them to discover novel solutions (Silver et al., 2016; Mankowitz et al., 2023), that a static human annotated dataset might not contain.

The recent success of reinforcement learning on real world problems makes it likely that RL techniques will become relevant for the broader machine learning audience. However, the field of RL currently has a large entry barrier, requiring a significant time investment to get started. Seminal work in the field (Schulman et al., 2015; 2016; Bellemare et al., 2017; Haarnoja et al., 2018a) often focuses on rigorous theoretical exposition and typically assumes that the reader is familiar with prior work. Existing textbooks (Sutton & Barto, 2018; François-Lavet et al., 2018) make little assumptions but are extensive in length. Our aim is to provide readers that are familiar with supervised machine learning an easy entry into the field of deep reinforcement learning to facilitate widespread adoption of these techniques. Towards this goal, we skip the typically rather lengthy introduction via tables and Markov decision processes. Instead, we introduce deep reinforcement learning through the intuitive lens of optimization. In only 25 pages, we introduce the reader to all relevant concepts to understand successful modern Deep RL algorithms like proximal policy optimization (PPO) (Schulman et al., 2017) or soft actor-critic (SAC) (Haarnoja et al., 2018a).

Our invitation to reinforcement learning is structured as follows. After discussing general notation in Section 2, we introduce reinforcement learning techniques by optimizing non-differentiable metrics in Section 3. We start with the standard supervised setting, e.g., image classification, assuming a fixed labeled dataset. This assumption is lifted in Section 4 where we discuss data collection in sequential decision making problems. In Section 5 and Section 6 we will extend the techniques from Section 3 to sequential decision making problems, such as robotic navigation. Fig. 1 provides a graphical representation of the content.

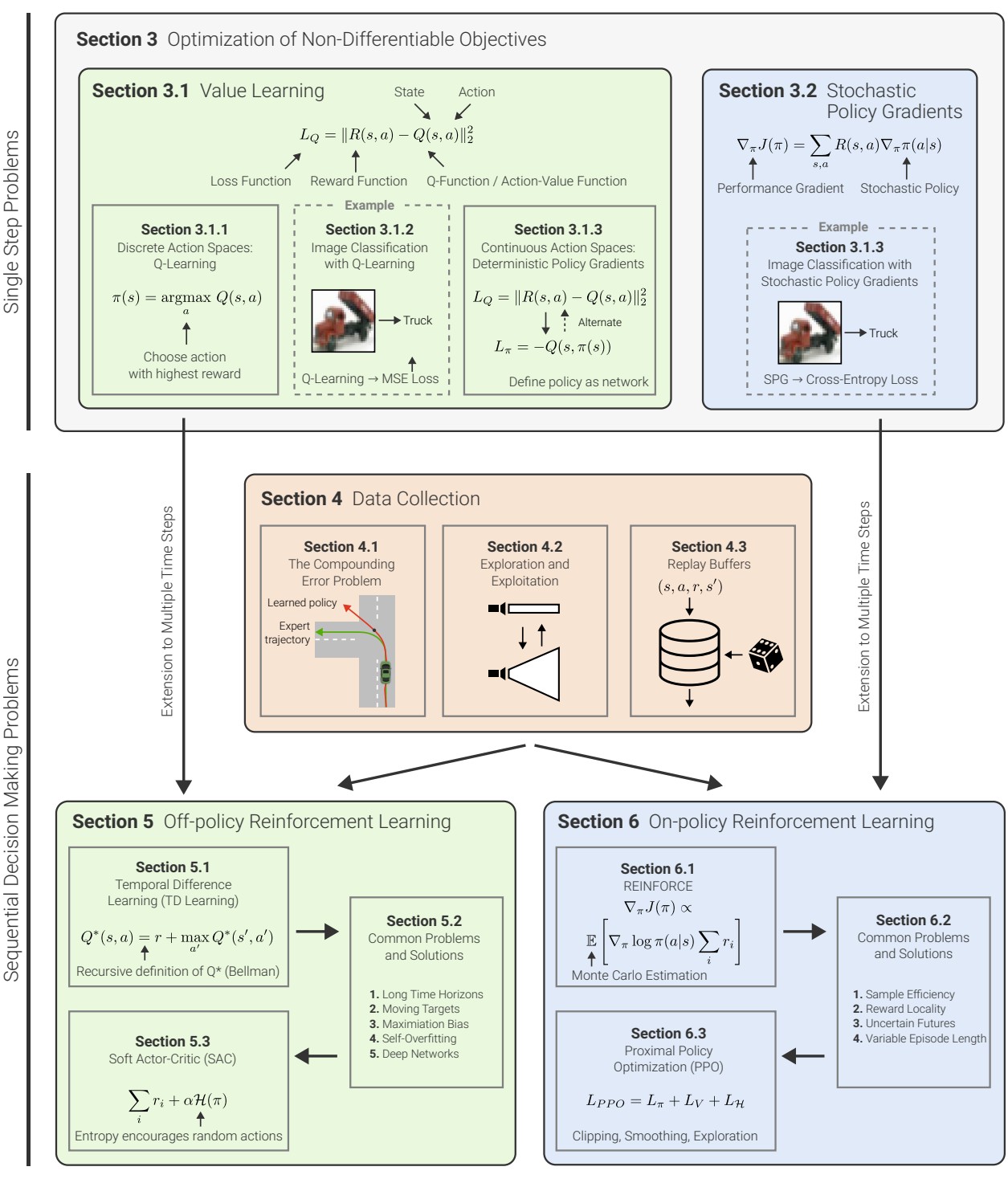

Figure 1: **An Invitation to Deep Reinforcement Learning.** This tutorial is structured as follows: We start by introducing reinforcement learning techniques through the lens of optimizing non-differentiable metrics for single step problems in Section 3. In particular, we discuss value learning in Section 3.1 and stochastic policy gradients in Section 3.2. For each category of algorithms, we provide a simple example assuming a fixed labeled dataset, thereby connecting RL to supervised learning objectives. This assumption is lifted in Section 4 where we discuss data collection for sequential decision making problems. Next, we extend the techniques from Section 3 to sequential (multi-step) decision making problems. More specifically, we extend value learning to off-policy RL in Section 5 and stochastic policy gradients to on-policy RL in Section 6. For both paradigms, we introduce basic learning algorithms (TD-Learning, REINFORCE), discuss common problems and solutions, and introduce a modern advanced algorithm (SAC, PPO).

## 2 Notation

In supervised learning (SL), the goal is to optimize a model to map inputs $x$ to correct predictions $y$. In the field of RL, the inputs $x$ are called the state $s$ and the predictions $y$ are called the actions $a$. The model is called the *policy* $\pi$ and maps states to actions $\pi(s) \to a$.

In this context, supervised learning *minimizes* a loss function $L(h(s), a) \to r$ where $h(s) \to a^\star$ is the function that maps the states $s$ (= input) to the label $a^\star$, treated as the optimal action (= ground truth label), and $r \in \mathbb{R}$ is a scalar loss value. The loss function, typically abbreviated as $L(a^\star, a)$, needs to be differentiable and is in most cases optimized using gradient-based methods based on samples drawn from a fixed dataset. The ground truth labels $a^\star$ are typically provided by human annotators.

In reinforcement learning, the scalar $r$ is called the *reward* and the objective is to *maximize* the *reward function* $R(s, a) \to r$. Reward functions $R$ represent a generalization of loss functions $L$ as they do not need to be differentiable and the optimal actions $a^\star = h(s)$ does not need to be known. It is important to remark that non-differentiable functions are not the primary limitation of supervised learning. Most loss functions like step-functions are differentiable almost everywhere. The main problem is that the gradients of step-functions are zero almost everywhere, and hence provide neither gradient direction nor gradient magnitude to the optimization algorithm. We use the term non-differentiable to also refer to objectives that are differentiable but whose gradient is zero almost everywhere. The reward function $R$ can be non-differentiable and the mathematical form of $R$ does not need to be known, as long as reward samples $r$ are available. Consider the execution speed of computer code as an example: we are able to measure runtime, but cannot compute it mathematically, as it depends on the physical properties of the hardware.

Like supervised learning, RL requires the objective to be *decomposable* which means that the objective can be computed for each individual state $s$. However, this is a fairly weak requirement. Objectives defined over entire datasets, for example mean average precision (mAP), are not decomposable but can be replaced with a decomposable reward function that strongly correlates with the original objective (Pinto et al., 2023).

The goal of policies $\pi(s) \to a$ is to predict actions that maximize the reward function $R$. Policies are typically deployed in so-called environments that will produce a next state $s'$ and alongside the reward $r$, after the model took an action. This process is repeated until a terminal state is reached which terminates the current *episode*. In classical supervised learning, the model makes only a single prediction, so the length of the episode is always one. When environments have more than one step, the goal is to maximize the *sum* of all rewards, which we will call the *return*. In multistep environments, rewards may depend on past states and actions. For $R(s, a) \to r$ to be well-defined, states are assumed to satisfy the *Markov property* which implies that they contain all relevant information about the past. In practice this is achieved by designing states to capture past information (Mnih et al., 2015) or by using recurrent neural networks (Bakker, 2001; Hausknecht & Stone, 2015; Narasimhan et al., 2015; Kapturowski et al., 2019) as a memory mechanism. Many works formalize the goal of maximizing return wrt. a particular reward function $R$ in an environment with states that have the Markov property as solving a *Markov Decision Process* (MDP). A Markov decision process is a 4-tuple $(S, A, P, R)$ containing the set of all states $S$, the set of all actions $A$, an environment transition kernel $P(s', s, a) = Pr(s_{t+1} = s' | s_t = s, a_t = a)$, describing the probability of the next state given the current state and action and a reward function $R$. While helpful for theoretical analysis we will not need the concept in the remainder of this tutorial.

In contrast to supervised learning, the data that the policy $\pi$ is trained with is typically collected during the training process by exploring the environment. Most reinforcement learning methods treat the problem of optimizing the reward function $R$ and the problem of data collection jointly. However, to understand the ideas behind RL algorithms, it is instructive to look at these problems individually. This will also help us to understand how the ideas behind RL can be applied to a much broader set of problems than the planning and control tasks they have originally been proposed for.

We follow the notation outlined in Table 1. Symbols will be introduced when they first become relevant, but not every time they are used. Table 1 can hence be used as a quick reference for reading later equations.

| Symbol | Explanation | Alternative |
|---|---|---|
| $s$ | state | input, observation, $x$ |
| $s'$ | next state | - |
| $\mathcal{S}$ | set of all states | dataset |
| $r$ | reward at current time step | objective, scalar loss |
| $r_i$ | reward at time step $i$ | - |
| $a$ | action | prediction, $y$ |
| $a'$ | next action | - |
| $a^\star$ | optimal action | - |
| $\mathcal{A}$ | set of all actions | - |
| $N$ | number of steps in an episode | - |
| $\pi(s)$ | policy | model |
| $\pi(a|s)$ | probabilistic policy | probabilistic model |
| $\pi_\beta$ | policy that collected the data | behavior policy |
| $\pi^\star$ | optimal policy | - |
| $Q(s,a)$ | action-value function, predicts expected return | critic, predicts expected reward in 1-step settings |
| $Q^\pi(s,a)$ | Q-function specific to $\pi$ | - |
| $Q'(s,a)$ | target Q-function | - |
| $h(s) \to a^\star$ | function mapping states to optimal actions | usually labels from human annotators |
| $R(s,a)$ | reward function | objective function |
| $L(a^\star,a)$ | supervised loss function | - |
| $\gamma$ | discount factor $\in [0,1]$ | - |
| $L_\pi$ | loss of policy | - |
| $\|\cdot\|_2^2$ | squared error | squared $l^2$-norm |
| $\theta$ | weights of a neural network | - |
| $J(\pi)$ | function measuring the performance of a policy | - |
| $\nabla_\pi$ | gradient wrt. the policy weights | - |
| $acc(s,a)$ | accuracy of class $a$ in state $s$ | - |
| $\mathcal{N}(a;\mu,\sigma)$ | PDF of the normal distribution | PDF of the Gaussian distribution |
| $\mathcal{H}(\pi(\cdot|s))$ | entropy of policy | - |
| $\alpha$ | hyperparameter in soft-actor critic | trades off entropy vs return |
| $\bar{\mathcal{H}}$ | hyperparameter in soft-actor critic | target entropy |
| G | objective to maximize | - |
| $a \sim \pi(\cdot|s)$ | sample drawn from distribution | - |
| $\xi$ | sample drawn from standard normal distribution | - |
| $\mathbb{E}$ | expectation | - |
| $\propto$ | proportional to | - |
| tanh | tangens hyperbolicus function | - |
| $D$ | replay buffer | - |
| $\tau$ | hyperparameter $\in [0,1]$ | speed of copying $Q$ to $Q'$ |
| $d^\pi(s)$ | probability of visiting state $s$ with policy $\pi$ | - |
| $b(s)$ | baseline, subtracted from return | - |
| $V^\pi(s)$ | value function, also referred to as critic | predicts expected return in state $s$ using policy $\pi$ |
| $G_n$ | $n$-step return | - |
| $G_\lambda$ | $\lambda$-return | eligibility trace |
| $A^\pi(s,a)$ | advantage function | - |
| $A^\pi_\lambda$ | generalized advantage function | advantage estimated with $\lambda$-return |
| $\psi$ | clipping hyperparameter in PPO | - |
| $\epsilon$ | percentage of random actions during data collection | - |
| $M$ | number of parallel actors in PPO | - |
| $B$ | temporary data buffer in PPO | - |

Table 1: **Notation.** Overview of the most commonly used symbols in this tutorial.

In this article, we only consider problems that are difficult enough to require function approximation. In particular, we assume that all functions are approximated by differentiable (deep) neural networks. In general, we try to keep equations readable by omitting indices when they can be inferred from the context. For example, optimizing a policy network $\pi$ shall be interpreted as optimizing the neural network weights $\theta$ that parameterize the policy $\pi_\theta$. The subscript of loss functions $L$ indicates which network is optimized. All loss functions are minimized. In cases where we want to maximize a term, we minimize the negative instead. We only consider problems with finite episode length for clarity. Most problems have finite episode length, but RL algorithms can also be extended to infinitely long episodes (Pardo et al., 2018; Sutton & Barto, 2018). To further simplify equations, we often omit what is called the discount factor $\gamma \in [0, 1]$. Optimizing for very long time horizons is hard, and practitioners often use discount factors to down-weight future rewards, limiting the effective time horizon that the algorithm optimizes for. This biases the solution, but may be necessary to improve convergence. We discuss discount factors in Section 6.2.2.

The remainder of this invitation to reinforcement learning is structured as illustrated in Fig. 1: In Section 3, we introduce reinforcement learning techniques by optimizing non-differentiable metrics. We start with the classic supervised prediction setting, e.g., image classification, assuming a fixed labeled dataset. This assumption is lifted in Section 4 where we discuss data collection in sequential decision making problems. In Section 5 and Section 6 we will extend the techniques from Section 3 to sequential decision making problems, such as robotic navigation.

## 3 Optimization of Non-Differentiable Objectives

To abstract away the complexity associated with sequential decision making problems, we start by considering environments of length one in this section (i.e., the policy only makes a single prediction) and assume that a labeled dataset is given. In the following, we will introduce the two most important techniques to maximize rewards without access to gradients of the reward function.

### 3.1 Value Learning

The most popular idea to maximize rewards without a gradient from the action $a$ to the reward $r$ is *value learning* (Sutton & Barto, 2018). The key idea of value learning is to directly predict the (expected) return rather than the action $a$ and to define the policy $\pi$ implicitly. Value learning simplifies to predicting the current reward $r$ in environments of length one, as there is no future reward. We generalize value learning to multi-step problems in Section 5. Optimization is carried out by minimizing the difference between the predicted reward and the observed reward using a regression loss.

In this family of approaches, an action-value function $Q$ predicts the reward $r$, given a state $s$ and action $a$:

$$Q(s, a) \rightarrow r \tag{1}$$

The goal of the Q-function is to predict the corresponding reward $r$ for *every* action $a$ given a state $s$, hence effectively approximating the underlying non-differentiable reward function $R(s, a)$. The Q-function is typically trained by minimizing a mean squared error (MSE) loss:

$$L_Q = (R(s, a) - Q(s, a))^2 \tag{2}$$

The policy $\pi$ is implicitly defined by choosing the action for which the Q-function predicts the highest reward:

$$\pi(s) = \operatorname*{argmax}_a Q(s, a) \tag{3}$$

#### 3.1.1 Discrete Action Spaces: Q-Learning

For problems with discrete actions, e.g. classification, this argmax can be evaluated by simply computing the Q-values for all actions as shown in Fig. 2a. Learning the Q-function is commonly referred to *Q-Learning* (Watkins, 1989; Watkins & Dayan, 1992) or *Deep Q-Learning* (Mnih et al., 2015) if the Q-function is a deep neural network. As evaluation of the argmax requires one forward pass per action $a \in |\mathcal{A}|$, this can become

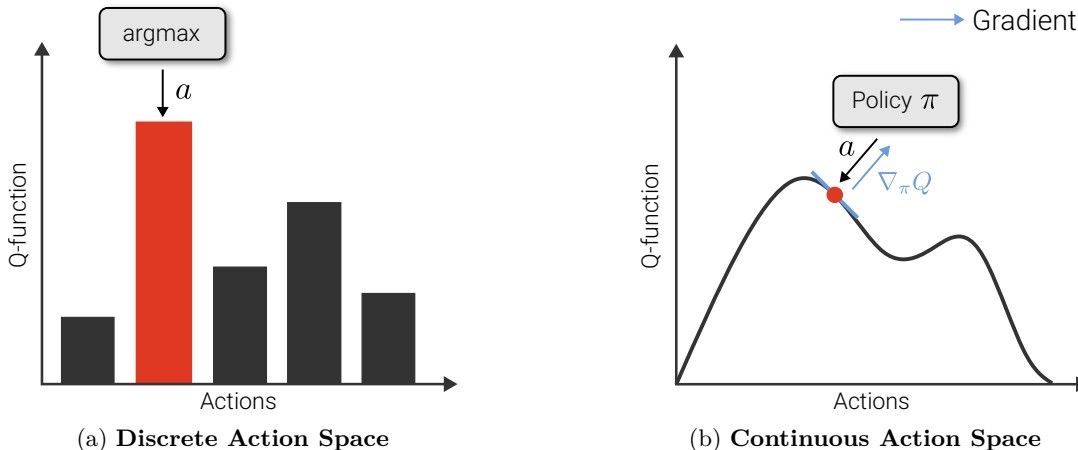

(a) **Discrete Action Space**      (b) **Continuous Action Space**

Figure 2: **Q-Functions.** We illustrate the predicted reward of a Q-function for a *fixed state*. (a) Discrete action space with 5 classes. The best action can be selected by computing all 5 Q-values (sequentially or in parallel). (b) 1-dimensional continuous action space. The maximum value cannot easily be found since the Q-function can only be evaluated at a finite amount of points. Instead, a policy network predicts the action with the highest reward. The policy is improved by following the gradient of the Q-function uphill.

inefficient for deep neural networks. In practice, Q-functions are typically defined to predict the reward for every action simultaneously. In this case, only one forward pass is required to select the best action:

$$L_Q = \left\| \left(r_1, r_2, \ldots, r_{|\mathcal{A}|}\right)^\top - Q(s) \right\|_2^2 \tag{4}$$

### 3.1.2 Example: Image Classification with Q-Learning

To demonstrate how Q-learning can be used to optimize a non-differentiable objective, we will use a simple image classification problem and the non-differentiable accuracy metric that is frequently used to measure the performance of classification models but that cannot directly be optimized using gradient-based optimization techniques. Using the optimization techniques of RL in combination with a static dataset is called *Offline RL*.

Our goal is to train a ResNet50 (He et al., 2016) for image classification on CIFAR-10 (Krizhevsky et al., 2009), optimizing accuracy directly with Q-learning. In this setting, the state is a 32x32 pixel image, the actions are the 10 class labels and the reward objective is the per class accuracy. As the action space is discrete, we use a ResNet50 (He et al., 2016) with 10 output nodes as our Q-function. The accuracy reward is 1 for the correct class and 0 otherwise. The reward labels can therefore be naturally represented as one-hot vectors, just like in supervised learning. Thus, changing the loss function from cross-entropy to mean squared error is the only change necessary to switch from supervised learning to Q-learning. We use the standard hyperparameters of the image classification library timm (version 0.9.7) (Wightman, 2019), except for a 10 times larger learning rate when training with the MSE loss. To classify an image, we select the class for which the Q-ResNet predicts the highest accuracy. We compare the MSE loss to the standard cross entropy (CE) loss without label smoothing on the CIFAR-10 dataset:

| Loss | Validation Accuracy ↑ |
|------|------|
| Cross-Entropy | 95.1 |
| Mean Squared Error | 95.4 |

As evident from the results above, Q-learning achieves similar accuracy to the cross-entropy (CE) loss. While it is well known that classification models can be trained with MSE (Hastie et al., 2009), here we

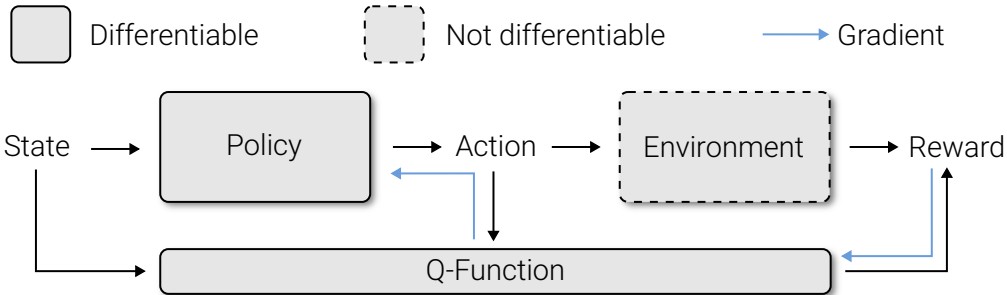

(a) **Q-learning** (here: actor-critic) bridges the non-differentiable environment by predicting the reward.

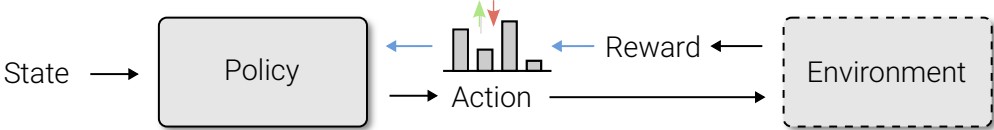

(b) **Policy gradient methods** up-weight actions that lead to high reward.

Figure 3: **Optimization of Non-Differentiable Objectives.** We compare Q-learning (continuous setting/actor-critic) in (a) to stochastic policy gradients in (b). Note that both Q-learning and stochastic policy gradients do not require differentiation of the environment.

show that this can be viewed as offline Q-learning and leads to policies that maximize accuracy. This result doesn't imply that Q-learning is competitive in every classification task. Q-learning can struggle to scale naively to high dimensional action spaces (de Wiele et al., 2020), which is important for classification benchmarks like ImageNet (Deng et al., 2009). It may be surpising that CE achieves similar accuracy on CIFAR-10, since it is motivated by information theory and considered a surrogate loss for accuracy (Song et al., 2016; Grabocka et al., 2019; Huang et al., 2021) However, as we will see in Section 3.2, the CE loss can be interpreted as another reinforcement learning technique (stochastic policy gradients).

### 3.1.3 Continuous Action Spaces: Deterministic Policy Gradients

When using Q-learning in settings with continuous action spaces, the problem arises that the argmax in Eq. (3) is intractable. A common solution to this problem is to define the policy explicitly as a neural network that predicts the argmax function. In this case, the policy $\pi$ is optimized to maximize the output of the Q-function with the following loss:

$$L_\pi = -Q\left(s, \pi(s)\right) \tag{5}$$

This technique, called the *deterministic policy gradient* (Silver et al., 2014; Lillicrap et al., 2016), is illustrated in Fig. 2b. One simply follows the gradient of the Q-function to find a local maximum. Optimization is performed by either first training the Q-function to convergence (given a fixed dataset) and afterwards the policy or by alternating between Eq. (2) and Eq. (5) which is more common in sequential decision making problems which typically require exploration to collect data. Algorithms that jointly learn a policy and a value function are sometimes called *actor-critic* methods (Konda & Tsitsiklis, 1999), where "actor" refers to the policy and "critic" describes the Q-function. The idea of using actor-critic learning is illustrated in Fig. 3a.

In the actor-critic approach, the Q-function is only used during training and discarded at inference time. This allows the Q-function to use privileged information such as labels or simulator access as input, which typically simplifies learning. In cases where the additional information is a label $a^\star$ the state often becomes redundant and can be removed, simplifying the Q-function to $Q(a^\star, a)$ and the policy loss to $L_\pi = -Q(a^\star, \pi(s))$. If the reward function $R$ is additionally differentiable, then there is no need to learn it with a Q-function, it can directly be used. Using the negative $L_2$ loss as reward function recovers supervised

regression: $L_\pi = L_2(a^\star, \pi(s))$. Hence, supervised regression can be viewed as a special case of actor-critic learning where the reward function is the negative $L_2$ distance to the ground truth label and all episodes have length one.

## 3.2 Stochastic Policy Gradients

The second popular idea to maximize non-differentiable rewards in reinforcement learning is called *stochastic policy gradients* or *policy gradients* in short. Policy gradient methods train a policy network $\pi(a|s)$ that predicts a probability distribution over actions. The central idea is to ignore the non-differentiable gap, and instead to use the rewards directly to change the action distribution. Action probabilities are up-weighted proportionally to the reward that they receive. Policy gradient methods require the output of the policy to be a probability distribution, such that up-weighting one action automatically down-weights all other actions. When this process is repeated over a large dataset, actions which achieve the highest rewards will get up weighted the most and hence will have the highest probability.

The idea of policy gradients is illustrated in Fig. 3b. Policy gradients are easy to derive from first principles in the supervised learning setting, which we will do in the following. Our goal is to optimize the neural network $\pi$ such that it maximizes the reward $r$ for every state $s$ in a training dataset $\mathcal{S}$. The dataset $\mathcal{S}$ consist of state-action pairs $(s, a) \in \mathcal{S}$ or state-action-reward triplets $(s, a, r) \in \mathcal{S}$ in case the reward function $R$ is unknown. The performance of the network $\pi$ is measured by the following function $J(\pi)$:

$$J(\pi) = \frac{1}{|S|} \sum_{(s,a) \in \mathcal{S}} R(s,a)\pi(a|s) \tag{6}$$

In other words, we maximize the expectation of the product of the reward for an action and the probability that the neural network $\pi$ would have taken that action over the empirical data distribution. For example, in the case where the reward is the accuracy of a class, the function $J(\pi)$ describes the training accuracy of the model. The function $J(\pi)$ has its global optimum at neural networks $\pi$ that put probability 1 on the action that attain the best reward for every state in the dataset. Note again, that the policy must be a probability distribution over actions to not yield degenerate solutions where the policy simply predicts $\infty$ for all positive rewards. The policy $\pi$ is optimized to maximize the performance $J(\pi)$ via standard gradient ascent by following its gradient:

$$\nabla_\pi J(\pi) = \frac{1}{|\mathcal{S}|} \sum_{(s,a) \in \mathcal{S}} R(s,a)\nabla_\pi \pi(a|s) \tag{7}$$

This is called the *policy gradient* which can be computed via the backpropagation algorithm as $\pi$ is differentiable. The non-differentiable reward function does not depend on the policy, hence it can be treated as a constant factor. In practice, the average gradient is computed over mini-batches and not the entire dataset.

To compute the policy gradient, we need to differentiate through the probability density function (PDF) of the probability distribution. With discrete actions, the categorical distribution is used. In settings with continuous action spaces, we require a continuous probability distribution, such as the Gaussian distribution for which the mean and standard deviation are predicted by the network: $\pi(a|s) = \mathcal{N}(a; \pi_\mu(s), \pi_\sigma(s))$. While policy gradients can optimize stochastic policies, the resulting uncertainty estimates are not necessarily well calibrated (Guo et al., 2017). During inference, the mean or argmax are often used for choosing an action.

### 3.2.1   Example: Image Classification with Stochastic Policy Gradients

Consider again the example of classification, but this time with policy gradients. The reward is accuracy $acc(s,a)$. The actions are classes and the accuracy is one, if $a$ matches the label $a^\star$ and zero otherwise:

$$\nabla_\pi J(\pi) = \frac{1}{|\mathcal{S}|} \sum_{(s,a)\in\mathcal{S}} acc(s,a)\nabla_\pi \pi(a|s) \tag{8}$$

Since all terms with zero accuracy cancel, we can simplify this equation as follows:

$$\nabla_\pi J(\pi) = \frac{1}{|\mathcal{S}|} \sum_{(s,a^\star)\in\mathcal{S}} \nabla_\pi \pi(a^\star|s) \tag{9}$$

Instead of maximizing $\pi(a^\star|s)$, one may choose to minimize the negative log probability $-\log \pi(a^\star|s)$, since the logarithm is a monotonic function and hence does not change the location of the global optimum. Doing so recovers the familiar cross-entropy loss which corresponds to the negative log-likelihood:

$$L_\pi = \underbrace{- \sum_{(s,a^\star)\in\mathcal{S}} \log \pi(a^\star|s)}_{\text{Cross Entropy}} = \underbrace{-\log \prod_{(s,a^\star)\in\mathcal{S}} \pi(a^\star|s)}_{\text{Negative Log-Likelihood}} \tag{10}$$

We see that accuracy maximization is another motivation for cross-entropy, besides the standard information theoretic derivation (Shore & Johnson, 1980) and maximum likelihood estimation (Bishop, 2006).

## 4   Data Collection

We have introduced the two most popular techniques to maximize non-differentiable rewards (value learning and policy gradients) in the supervised setting where we have a dataset and only make a single prediction. Next, we will introduce extensions of these ideas to sequential decision making problems. These type of settings introduce additional challenges for data collection that we will discuss in this section. In Section 5 we will extend value learning to sequential decision making problems. For policy gradients, this extension introduces a dependency on the data collection, hence we will cover those challenges separately in Section 6.

The standard machine learning setting starts with a given dataset and tries to find the best model for that dataset. When using reinforcement learning for sequential decision making tasks, data collection is typically considered part of the problem as data must be collected through interaction with the environment.

### 4.1   The Compounding Error Problem

A typical assumption that neural networks require for generalization is that the data they are trained with covers the underlying distribution well. The dataset is assumed to contain independent and identically distributed (IID) samples from the data generating distribution. Validation sets used to evaluate a model often satisfy this assumption because they are randomly sampled subsets of the whole dataset. In the IID setting, neural networks typically work well. However, neural networks are known to struggle with Non-IID data which is very different from the training data (Beery et al., 2018; Schölkopf, 2022).

In sequential decision making problems, it is very hard to construct datasets whose samples are IID because of the feedback loop. During inference, the next state a neural network observes depends on the actions it has predicted in past states. Since the network might take different actions than an annotator that collected the dataset, it may visit states very different from those present in the dataset, i.e. the network will encounter out-of-distribution data at test time. Consider for example the problem of autonomous driving. A human annotator may collect a dataset by driving along various routes. As the data collectors are expert drivers, they will always drive near the center of lanes. A neural network trained with such data may make mistakes at test time, deviating from the center of the lane. Its new inputs will now be out-of-distribution, because the human expert drivers did not make such a mistake, typically leading to even worse predictions, *compounding*

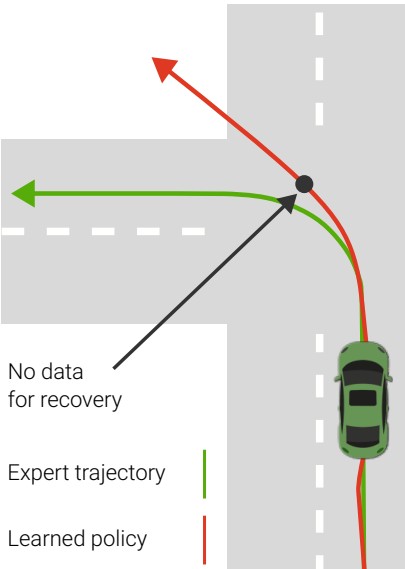

Figure 4: **Compounding Error Problem.** Small mistakes lead to Non-IID states that increase the error.

*errors*, until the policy catastrophically fails. In other words, the training data is missing examples of how to recover from mistakes that an annotator does not make.

This problem is illustrated in Fig. 4 where the learned red policy deviates from the collected training data in blue during the turn, eventually leading to catastrophic failure. The compounding error problem is an important reason why offline RL is difficult to apply to sequential decision making problems (Levine et al., 2020). The typical solution to the compounding error problem is to let the network collect its own data (Ross et al., 2011) avoiding this distribution shift. This is also called *online learning*. As a result, the training data contains most states that the network might reach, avoiding out-of-distribution data. Since data collection is part of the training loop in online learning, automatic annotation or computation of rewards is important for efficiency. Training in simulations is the most common solution used to achieve this efficiency. Collecting data during training introduces additional challenges that we will discuss in the following.

## 4.2 Exploration and Exploitation

To automatically collect data during training, the policy is deployed in an environment and the observed states, actions, next states and rewards are saved for training. The policy is then trained with the available data, alternating between collection and training. To increase efficiency, it is customary to run several models in parallel (Mnih et al., 2016; Horgan et al., 2018; Espeholt et al., 2018).

By picking the best currently known action, the agent is following the current policy during data collection. This behavior is called *exploitation*. However, to discover new states, other actions must sometimes be chosen. This is called *exploration*. In settings with discrete action spaces, exploration can be performed by picking a random action with a certain probability $\epsilon$ and the best known action otherwise. This strategy is called $\epsilon$-*greedy*. The $\epsilon$ parameter regulates the *exploration-exploitation* tradeoff. Additionally, the $\epsilon$ hyperparameter can be decayed towards zero over the course of training. In settings with continuous action spaces, noise can be added to the predicted action instead. Typically, Gaussian noise with mean zero and a standard deviation that is a hyperparameter analogously to the $\epsilon$ value (Eberhard et al., 2023) is used.

In environments where random actions induce catastrophic failure, performing random actions may prevent the agent from reaching states far away from the start, rendering data collection with $\epsilon$-greedy brittle. One approach to mitigate this problem is to learn the amount of noise to be applied. With policies that output a probability distribution, the action can be sampled from this distribution. Algorithms using this idea typically add an objective to increase the entropy of the policy to their training objective, so that the

network is encouraged to explore but only if exploration doesn't impact the expected performance too much (Schulman et al., 2017; Haarnoja et al., 2018a;b). For tasks with deterministic outputs, a learnable amount of Gaussian noise can be added to the hidden units in the linear layers instead (Fortunato et al., 2018; Hessel et al., 2018).

The exploration-exploitation tradeoff may be somewhat of a misleading name in the context of RL. Unlike animals, that need to learn at test time, RL agents typically have a training phase where the agent is trying to learn and only the final performance matters. At test time, the trained agent does not learn anymore and only exploits the best known option. Since the rewards that the agent obtains during training typically do not matter to the designer of the algorithm, it may seem beneficial to only explore and never exploit such that the policy can learn as much as possible. This, however, is not a good strategy because the agent is unlikely to reach interesting states far away from the start while only performing random actions. The purpose of exploitation during training is to *return* to previously visited *promising* states, such that the network can explore from relevant states different from the initial state. Hence, it might be better to speak of the *exploration-return* tradeoff in the context of RL. Instead of using exploitation to return, one may also choose to simply reset a simulator to a particular state. Ecoffet et al. (2021) has used this idea by keeping a buffer of all visited states. They collected data by simply resetting the simulator to previously visited states and performing random exploration. This approach achieved above human level results on all Atari games (Bellemare et al., 2013), which is an important RL benchmark. However, keeping a buffer of all visited states requires strong compression and might not scale to more complex environments.

A particularly difficult challenge for data collection is posed by environments where random actions will not lead to states with different rewards, i.e., because rewards are sparse. For example, consider an environment where the agent is playing chess against a grand-master level chess bot and the reward is +1 for a win, -1 for a loss and 0 otherwise. A policy collecting data by performing random actions is extremely unlikely to win, and hence all collected data will have the same reward of -1. Reward optimization is prevented because no data with high rewards will be collected, all actions seem equally bad to the optimizer. This is also the case in this example if the agent samples an improved action, since the optimizer will not know that the action was better unless it ultimately led to victory. Various solutions to the problem of sparse rewards have been proposed. One integral technique in chess and other board games is called *self-play* (Samuel, 1959; Tesauro, 1995; Silver et al., 2017). Instead of playing against a grand-master level opponent from the start, the policy is playing against itself or a past version of itself. This has the effect that the difficulty of the task is automatically adjusted to the skill of the current policy. Self-play ensures that both victory and losses are observed, and winning automatically becomes harder as the policy improves. Another option is to incorporate prior knowledge, for example in the form of human expert demonstrations (Silver et al., 2016; Vecerík et al., 2017; Chekroun et al., 2021). Lastly, some approaches maximize different rewards that make optimization easier and guide data collection. Examples include shaped rewards (Ng et al., 1999) which are reward functions that are engineered for that purpose or *intrinsic motivation* (Aubret et al., 2019) which are objectives that guide data collection towards states that novel or "surprising". While there are some solutions for particular environments, optimizing environments with sparse rewards is still a subject of ongoing research (Chen et al., 2022).

Another idea to make sparse reward settings or more generally hard tasks easier to learn is *curriculum learning* (Selfridge et al., 1985; Krueger & Dayan, 2009; Bengio et al., 2009). In curriculum learning, the environment or data is changed over time, typically increasing the difficulty of the task. The goal is that by learning the easier task first, the policy develops abilities that help it solve the harder task that appear later during training. Curriculum learning can aid data collection when the agent, by solving the easier environments, acquires the ability to reach promising states in the harder environments, that it would not have reached if it started from scratch in the harder environment. Additionally, curriculum learning was also empirically observed to improve optimization (Bengio et al., 2009). Designing a curriculum of increasingly harder environments requires domain knowledge and engineering but can be a crucial technique to further improve the performance of state-of-the-art methods. An example would be legged locomotion (Lee et al., 2020; Miki et al., 2022), a task where a legged robot needs to learn how to walk, where the difficulty of the terrain as well as external disturbances applied to the robot are gradually increased, adapting the difficulty of the environment to the ability of the policy. Automating the design of curricula is currently an active area of

research (Leibo et al., 2019; Portelas et al., 2020; Dennis et al., 2020; Jiang et al., 2021; Parker-Holder et al., 2022; Li et al., 2024) called *automatic curriculum learning*. Self-play, introduced in the last paragraph, can also be viewed as an automatic curriculum learning method, because it can be used without much domain knowledge in environments that feature competitive play.

### 4.3 Replay Buffers

When using Q-learning, data is typically stored in replay buffers (Lin, 1992; Mnih et al., 2015). Replay buffers are first-in-first-out queues with a fixed size. Consequently, during training, old data is eventually discarded. This keeps memory requirements of reinforcement learning constant wrt. to training time. During training, data is uniformly sampled from the replay buffer. This has the effect of breaking correlations between samples because they are selected from different episodes.

However, datasets in RL are typically quite biased (Nikishin et al., 2022). Initially, the bias is towards states around the starting region and later towards high reward trajectories, in particular if $\epsilon$ is decayed. Uniformly sampling data from the replay buffer is suboptimal because some samples are more informative (less redundant) than others. Prioritized experience replay (Schaul et al., 2015) addresses this issue by sampling data points proportionally to their loss when they were used the last time. This measures which samples are not well-fitted yet. New samples are given the maximum priority to make sure they are used at least once. Since prioritized experience replay automatically adjusts the data distribution based on the prediction capabilities of the model it can also be viewed as an automatic curriculum learning method (Portelas et al., 2020). Today, prioritized replay buffers are part of the standard RL toolset and used in various Q-learning based methods (Hessel et al., 2018; Badia et al., 2020).

## 5 Off-Policy Reinforcement Learning

We are now going to extend the idea of *value learning* presented in Section 3.1 to sequential decision making problems. In this setting, the environment generates a next state $s'$ in addition to the reward, that is fed back into the policy $\pi$. This process is repeated until a terminal state is reached, finishing the episode. Data is collected by the policy of the current training iteration and stored in a replay buffer as discussed in Section 4. The replay buffer consists of data collected by different policies because the policy is updated constantly during training. Algorithms that allow training with data collected by policies that are different from the current policy are called *off-policy*. This is in contrast to *on-policy* learning, described in Section 6, where data is assumed to be collected by the policy of the current training iteration. The advantage of off-policy learning is that it enables more flexible data collection strategies and increases sample efficiency as it reuses collected data and hence requires fewer interactions with the environment.

### 5.1 Temporal Difference Learning (TD Learning)

In sequential decision making problems, the objective is to maximize the *sum of rewards* called the *return*. The naive extension of Q-Learning therefore is to predict the return via the Q-function:

$$L_{Q_{\pi_\beta}} = \left( \sum_{i=t}^{N} r_i - Q^{\pi_\beta}(s, a) \right)^2 \tag{11}$$

Here, $N$ is the number of future environment steps in an episode, $r_i$ denotes the reward at time step $i$ and $t$ is the current time step. The return is called the *Monte Carlo* objective (Sutton & Barto, 2018). A difficulty in sequential problems is that the weights of the Q-function are dependent on a particular policy $\pi$. Here, $\pi$ is the policy that collected the data, called the *behavior policy* $\pi_\beta$. The definition of the Q-function is to predict the expected return when taking action $a$ in state $s$ and taking the actions predicted by the policy $\pi_\beta$ in all future states. The first reward $r_t$ is independent of $\pi_\beta$ because we condition on the first action as input. Future actions are not conditioned on, and hence the neural network weights of the Q-function need to be specific to a particular policy $\pi$ that in future states will take the actions that lead to the observed rewards. In other words, the learned Q-function becomes specific to a particular policy, which is why we change our notation to $Q^{\pi_\beta}$.

However, we ideally like to obtain the Q-function of the optimal policy $\pi^\star$, not the behavior policy $\pi_\beta$. The Q-function of the optimal policy can be defined recursively via the *Bellman equation* (Bellman & Dreyfus, 1962; Sutton & Barto, 2018):

$$Q^{\pi^\star}(s,a) = r + \max_{a'} Q^{\pi^\star}(s',a') \tag{12}$$

Here, $s'$ is the state that follows $s$ when taking action $a$ and $r$ denotes the *current* reward[1]. Exploiting the Bellman equation, TD-Learning uses the right side of Eq. (12) as target for a learned Q-function:

$$L_Q = \left(r + \max_{a'} Q(s',a') - Q(s,a)\right)^2 \tag{13}$$

The current reward $r = R(s,a)$ only depends on the inputs of the Q-function, $s$ and $a$. By using the max operator, we choose the optimal action $a$ that maximizes the predicted reward, thereby removing the dependency on the data collecting policy $\pi_\beta$. Hence, TD-Learning can be used *off-policy*.

To train with Eq. (13), data is collected by the policy of the current training iteration and stored as $(s, a, r, s')$ quadruples in a replay buffer, as discussed in Section 4. The replay buffer allows us to reuse data collected by policies from earlier training iterations. Note that the term $\max_{a'} Q(s',a')$ changes when we update the Q-function during training. When collected data samples are reused, the label will therefore be recomputed. Note, that the Q-function in the $\max_{a'} Q(s',a')$ term is typically treated as a constant and not differentiated, which is why this is called a *semi-gradient* method (Sutton & Barto, 2018).

For settings with continuous action spaces, the max operation is again intractable. As before, the solution is to predict the action $a'$ with a policy network $\pi(s)$:

$$L_Q = (r + Q(s', \pi(s')) - Q(s,a))^2 \tag{14}$$

and to update the policy by alternating between Eq. (14) and the deterministic policy gradient from Eq. (5).

This form of training is called *off-policy learning* because the policy $\pi_\beta$ that collected the data is different from the policy $\pi$ that we are currently training. Off-policy learning is appealing because the reuse of samples reduces the amount of environment interactions needed to train the policy. Optimizing Eq. (13) is also called *temporal difference learning*, or *TD(0)*, because we optimize a function by minimizing the difference between its current and next prediction. It is well known that optimizing Eq. (13) converges to the optimal solution for simple problems where the Q-function is represented by a table (Watkins & Dayan, 1992; Tsitsiklis, 1994). This guarantee does not hold when using function approximation (Watkins, 1989) but with the right techniques it is also possible to successfully train Q-functions represented by neural networks (Mnih et al., 2015).

## 5.2 Common Problems and Solutions

We now discuss some of the problems that arise when training deep Q-networks as well as possible solutions.

### 5.2.1 Long Time Horizons

Optimizing for long time horizons is in general challenging, as the future is typically hard to predict. The same state and action pair can lead to different returns as the future is typically stochastic. Observed returns can therefore have high variance. A common technique to reduce this variance is by limiting the effective time horizon of the optimization by applying a discount factor $\gamma \in [0,1]$ to future rewards:

$$L_Q = (r + \gamma Q(s', \pi(s')) - Q(s,a))^2 \tag{15}$$

Due to the recursive nature of the Bellman equation, future rewards will at some point implicitly be multiplied by almost 0 when using a discount factor $\gamma < 1$. This softly limits the effective time horizon. Doing so biases the objective, but is often used in practice to improve convergence. Limiting the effective time horizon with discount factors is a general idea that can be used with almost any RL algorithm. For clarity, we will omit discount factors from equations, except when presenting concrete algorithms.

---

[1]For clarity, we have omitted the termination condition, that defines $Q^{\pi^\star}(s,a) = r$ for the last state of an episode.

### 5.2.2 Moving Targets

The target we are optimizing for is constantly changing and hence may lead to oscillations or even divergence. Therefore, a so-called target network $Q'$ (Mnih et al., 2015) is often used as target. $Q'$ is a copy of the primary Q-network that is only copied once in a while instead of every iteration, stabilizing the training objective:

$$L_Q = \left( r + \max_{a'} Q'(s', a') - Q(s, a) \right)^2 \tag{16}$$

### 5.2.3 Maximization Bias

A learned Q-function is a noisy predictor for the actual return. As noisy estimates will sometimes be too large, the max operation in Eq. (16) will lead to a *maximization bias* (Thrun & Schwartz, 1993). This problem is typically addressed using the idea of *double Q-learning* (Hasselt, 2010; Hasselt et al., 2016). While double Q-learning also leads to a biased estimator, it has a negative bias rather than a positive bias as with regular Q-learning, empirically leading to better results. The core idea behind double Q-Learning is to use two different Q-functions, one ($Q$) for choosing the action and one ($Q'$) for evaluating it:

$$L_Q = \left( r + Q' \left( s', \operatorname*{argmax}_{a'} Q(s', a') \right) - Q(s, a) \right)^2 \tag{17}$$

The policy induced by the Q-function changes quickly in settings with discrete action spaces (Schaul et al., 2022), which enables the use of the target network $Q'$ as the second network. In settings with continuous actions, the policy is observed to change more slowly (Fujimoto et al., 2018), hence $Q$ and $Q'$ are too similar. Two separate Q-networks ($Q_1, Q_2$) are therefore trained and the minimum between them is used which also reduces the maximization bias:

$$L_{Q_i} = \left( r + \min_{i=1,2} Q_i'(s', \pi(s')) - Q_i(s, a) \right)^2 \tag{18}$$

### 5.2.4 Self-Overfitting

The learned Q-function in Eq. (13) has control over both the label and the prediction. During training, it can learn to abuse this power by "collapsing" to a constant value. This is called self-overfitting (Cetin et al., 2022) and is similar to the collapsing problem in self-supervised learning (Chen & He, 2021). The loss will then be equivalent to the squared reward as the terms involving the Q-functions will vanish. The return is typically much larger than the current reward which gives the collapsed solution a relatively good loss. A simple but effective empirical strategy to mitigate self-overfitting in visual domains is to use shift augmentations for the input images (Laskin et al., 2020; Yarats et al., 2021; 2022; Cetin et al., 2022).

The collapsed solution has a particularly low loss if rewards are sparse (i.e., if they are mostly 0). This can be mitigated by using the next $n$ rewards instead of just the next reward to optimize the Q-function, which increases the reward density in the target but increases the variance in the objective because future rewards are uncertain. This is called the *n-step return* (Watkins, 1989; Peng & Williams, 1996):

$$L_Q = \left( \sum_{i=t}^{t+n-1} r_i + Q' \left( s_{t+n}, \operatorname*{argmax}_{a_{t+n}} Q(s_{t+n}, a_{t+n}) \right) - Q(s_t, a_t) \right)^2 \tag{19}$$

N-step returns are a general idea that can be employed with any algorithm that learns a value/Q-function. The hyperparameter $n$ represents a trade-off, where the bias coming from incorrectly estimated target Q-function decreases, and the variance coming from the stochastic nature of future rewards increases, with higher values of $n$. Another way to address the problem of sparse rewards is to maximize a different reward function that provides dense feedback at every time step. Designing this reward function is called *reward shaping* (Ng et al., 1999). Optimizing for shaped rewards can lead to suboptimal policies wrt. to the original reward function, but in practice often results in better policies because shaped reward functions are designed to ease optimization. Shaping rewards is independent of the learning algorithm and hence can be used with any RL algorithm. When designing a shaped reward function, particular care has to be taken to not introduce "loopholes", where simple undesired behaviors lead to high rewards (Clark & Amodei, 2016). Neural networks can learn to exploit such loopholes, which is called *shortcut learning* (Geirhos et al., 2020).

### 5.2.5 Deep Networks

Off-policy Q-learning methods often struggle to optimize very deep networks (Bjorck et al., 2021). Most architectures used in these type of methods only consists of a few layers and have simple visual or privileged inputs. Some success was reported in training a deep architecture without normalization layers (Kapturowski et al., 2023) but in general this problem is not fully understood yet and an active area of research (Schwarzer et al., 2023). To apply value learning to settings with high dimensional states, such as images, practitioners typically pre-train the networks (Liang et al., 2018), or predict low dimensional representations of the state (Toromanoff et al., 2020) with supervised learning.

Many algorithms exist that extend the basic Deep Q-learning algorithm (Schaul et al., 2015; Hasselt et al., 2016; Fujimoto et al., 2018; Hessel et al., 2018; Kapturowski et al., 2019; Badia et al., 2020; Kapturowski et al., 2023). In the next section, we will cover Soft Actor-Critic as a concrete example. Soft Actor-Critic is a popular algorithm based on Q-learning that incorporates many of the ideas discussed in this section into a single algorithm.

### 5.3 Soft Actor-Critic (SAC)

Soft Actor-Critic (Haarnoja et al., 2018a;b) is a popular off-policy actor-critic algorithm for continuous control where the policy $\pi$ is a Gaussian probability distribution. Unlike other actor-critic algorithms, it is comparably stable with respect to its hyperparameters, often working "out of the box" with default values.

Soft Actor-Critic maximizes the return while encouraging random behavior for better exploration. The randomness of the policy $\pi$ is measured via the entropy $\mathcal{H}$ over its actions $a$:

$$\sum_i^N r_i + \alpha \mathcal{H}(\pi(\cdot|s_i)) \tag{20}$$

The temperature parameter $\alpha$ is a hyperparameter defining the tradeoff between exploration and performance and $\mathcal{H}(\pi(\cdot|s)) = \mathbb{E}_{a\sim\pi}[-\log \pi(a|s)]$. Besides balancing exploration, a second advantage is that the network is encouraged to put equal probability on equally good actions. As is typical in value learning, the goal is to predict this objective using a Q-function. Soft Actor-Critic trains two $Q$-functions to address the maximization bias, as introduced in Section 5.2.3. It also uses target networks $Q'$ as objective to stabilize training, as discussed in Section 5.2.2. The Q-functions are trained with temporal difference learning with the following objective, where $a'$ is a sample from the probabilistic policy:

$$L_{Q_i} = \left( r + \min_{i=1,2} Q'_i(s', a') - \alpha \, \log \pi(a'|s') - Q_i(s, a) \right)^2 \qquad a' \sim \pi(\cdot|s') \tag{21}$$

The policy is updated analogously with the deterministic policy gradient.

$$L_\pi = -\min_{i=1,2} Q_i(s, a) + \alpha \, \log \pi(a|s), \qquad a \sim \pi(\cdot|s) \tag{22}$$

Backpropagating the gradients of this loss to the policy involves differentiating through a sample from a probability distribution, which in general is not possible. For some simple distributions like the Gaussian distribution, however, gradient computation is indeed possible by using the so called *reparametrization trick* (Kingma & Welling, 2014). Due to special properties of the Gaussian distribution, the following two action samples come from the same distribution:

$$a \sim \mathcal{N}\left(\pi_\mu(s), \pi_\sigma(s)\right) \tag{23}$$

$$a \sim \pi_\mu(s) + \pi_\sigma(s)\,\xi, \ \ \xi \sim \mathcal{N}(0, 1) \tag{24}$$

Eq. (24) is differentiable because the sample $\xi$ does not depend on the network and can be treated as a constant during backpropagation. The Soft Actor-Critic model uses the Gaussian distribution due to this property. Samples from a Gaussian distribution have full support, i.e., they lie in the range $[-\infty,$

$\infty$]. However, actions are often defined within an interval. The Soft Actor-Critic algorithm uses the tanh activation function to map actions into the range [-1,1]. Samples from the policy are therefore computed via:

$$\pi(s) = \tanh\left(\pi_\mu(s) + \pi_\sigma(s)\,\xi\right), \quad \xi \sim \mathcal{N}(0,1) \tag{25}$$

Adding a tanh activation function requires adjusting the PDF $\pi(a|s)$, however, an analytical solution exists. Since we have a stochastic policy that is encouraged to explore through its training objective, we can use the same policy for collecting training data by sampling from its distribution. The data is stored in a replay buffer as discussed in Section 4.3. It is important to tune the temperature parameter $\alpha$ per environment. In practice $\alpha$ can be tuned automatically during training (Haarnoja et al., 2018b), using the following loss:

$$L_\alpha = -\alpha\,\log\pi(a|s) - \alpha\,\bar{\mathcal{H}} \qquad a \sim \pi(\cdot|s) \tag{26}$$

Tuning $\alpha$ introduces a new hyperparameter $\bar{\mathcal{H}}$. However, $\bar{\mathcal{H}}$ is empirically robust across environments and is frequently chosen as the negative number of action dimensions (Haarnoja et al., 2018b).

---

**Algorithm 1** Soft Actor-Critic

---
Create empty replay buffer $\mathcal{D}$
**for** iterations **do**                                                        $\triangleright$ Training loop
    Collect an episode of data with $\pi$
    Store collected data in replay buffer $\mathcal{D}$
    Sample minibatch $(s, a, r, s') \sim \mathcal{D}$
    Update Q functions $L_{Q_i} = \left(r + \gamma\min_{i=1,2} Q_i'(s', a') - \alpha\,\log\pi(a'|s') - Q_i(s, a)\right)^2 \qquad a' \sim \pi(\cdot|s')$
    Update policy $L_\pi = -\min_{i=1,2} Q_i(s, a) + \alpha\,\log\pi(a|s) \qquad a \sim \pi(\cdot|s)$
    Update temperature $L_\alpha = -\alpha\,\log\pi(a|s) - \alpha\,\bar{\mathcal{H}} \qquad a \sim \pi(\cdot|s)$
    Update target network $Q' \leftarrow \tau Q' + (1 - \tau)Q$
**end for**

---

Algorithm 1 describes the Soft Actor-Critic algorithm. Soft Actor-Critic uses discount factors as discussed in Section 5.2.1. Target networks are smoothly updated via a running average with hyperparameter $\tau$. The Soft Actor-Critic algorithm has been empirically found to be robust to hyperparameter choices. It works well on problems like control in robotics (Haarnoja et al., 2018b) where state spaces are low dimensional and small neural networks are sufficient. With some extensions, it has also seen remarkable success in autonomous racing (Wurman et al., 2022). When using CNN encoders in visual domains, standard Soft Actor-Critc easily overfits, leading to poor performance. However, when using data augmentation and $n$-step returns, as discussed in Section 5.2.4, the algorithm can also be applied to these domains (Laskin et al., 2020; Yarats et al., 2021; 2022; Cetin et al., 2022). Soft Actor-Critic can be trained off-policy due to the recursive formulation of the Q-function, which means samples from the replay buffer $\mathcal{D}$ can be reused multiple times, reducing the amount of environment interactions required for convergence.

# 6 On-policy Reinforcement Learning

We are now going to extend the idea of *stochastic policy gradients* presented in Section 3.2 to sequential decision making problems. The objective in this setting, the sum of rewards, depends on which actions the policy will predict in future states, which makes computing the policy gradient difficult. Fortunately, it is still possible to compute the policy gradient in sequential settings when the data is collected *on-policy*. On-policy RL describes the setting where the data collecting policy $\pi_\beta$ is the same as the training policy $\pi$. This is achieved by discarding all collected data after every training iteration and collecting new data with the updated policy. On-policy algorithms therefore do not use replay buffers, but store data in temporary buffers that are cleared at the start of the next iteration of the algorithm. In principle, it is also possible to use Q-learning in on-policy settings, but in practice this is rarely done because policy gradient methods empirically achieve better performance in on-policy settings (Mnih et al., 2016). On-policy learning is challenging because there are strong correlations between sequentially collected samples, and rare examples can only be used once. However, on-policy learning with stochastic policy gradients, unlike Q-learning, guarantees

convergence to a local minimum also for deep neural networks (Wang et al., 2020b). In the following, we will first introduce the vanilla on-policy policy gradient algorithm *REINFORCE*. REINFORCE has a number of drawbacks, such as sample inefficiency and high variance. We will hence also discuss ways to mitigate these problems. Finally, we will introduce proximal policy optimization (PPO), a state-of-the-art algorithm that combines these ideas into a single method.

## 6.1 REINFORCE

The central idea of policy gradients is to upweight action probabilities proportional to the reward they receive, as introduced in Section 3.2. This idea can be extended to multistep environments by upweighting actions proportional to the return (sum of rewards). For an intuitive example, consider a chess policy, where the reward is 1 in case of victory, -1 in case of defeat and 0 otherwise. Policy gradients upweight actions that led to victory and down weights actions that led to defeat. Doing so maximizes the obtained return in expectation. Algorithms using this idea are called *REINFORCE* (Williams, 1992) algorithms, although we will use this name only for the vanilla policy gradient algorithm introduced in this section. REINFORCE can be derived similarly to the derivation in Section 3.2 by defining the performance function $J$ of the neural network $\pi$. An additional difficulty in sequential decision making is that the state and actions are not sampled uniformly from a dataset anymore. Instead, data is actively collected, hence one needs to consider the probability of visiting a state (from possible start states) given a policy. We denote this probability as $d^\pi(s)$ and define $\mathcal{S}$ as the set of all possible states here. The value function $V^\pi(s)$ describes the expected return in state $s$ when following policy $\pi$. It can be used to define the performance $J$ by computing the value from the start state $s_0$:

$$J(\pi) = V^\pi(s_0) \tag{27}$$

The goal is now to maximize $J$ by computing its gradient $\nabla_\pi J(\pi)$. Due to a result called the *policy gradient theorem* (Sutton et al., 1999; Marbach & Tsitsiklis, 2001) this gradient can be written in the following form:

$$\nabla_\pi J(\pi) \propto \sum_{s \in \mathcal{S}} d^\pi(s) \sum_{a \in \mathcal{A}} \left( \nabla_\pi \pi(a|s) \right) Q^\pi(s, a) \tag{28}$$

Here, the factor of proportionality is the average length of an episode (Sutton & Barto, 2018). For a derivation of the policy gradient theorem, see Sutton & Barto (2018) or Levine (2023). The policy gradient theorem is a remarkable result because it shows that $\nabla_\pi J$ can be computed without backpropagating through the state distribution $d^\pi$ or the action-value function $Q^\pi$ which makes it feasible to compute this gradient in practice. Additionally, the theorem also applies when $\pi$ is a deep neural network (Wang et al., 2020b). It is important to note that, unlike in value learning, the Q-function in Eq. (28) is the true Q-function, describing the expected return when taking action $a$ in state $s$ and following the policy $\pi$ afterwards. We use parentheses to emphasize that the Q-function is a constant in the gradient computation. When training with on-policy data, the frequency of the actions we observe are also dependent on the probability of the policy taking that action. This can be corrected for by using the log derivative trick $\nabla \log f(x) = \frac{\nabla f(x)}{x}$:

$$
\begin{aligned}
\nabla_\pi J(\pi) &\propto \sum_{s \in \mathcal{S}} d^\pi(s) \sum_{a \in \mathcal{A}} \frac{\pi(a|s)}{\pi(a|s)} (\nabla_\pi \pi(a|s)) Q^\pi(s, a) \\
&= \sum_{s \in \mathcal{S}} d^\pi(s) \sum_{a \in \mathcal{A}} \pi(a|s) (\nabla_\pi \log \pi(a|s)) Q^\pi(s, a) \\
&= \mathbb{E}_{s \sim d^\pi(s), a \sim \pi(\cdot|s)} [(\nabla_\pi \log \pi(a|s)) Q^\pi(s, a)]
\end{aligned}
\tag{29}
$$

The sums now represent an expectation and samples from that expectation are equivalent to on-policy collected data because the sum over states is weighted by how likely the policy will visit them, and the sum over actions is weighted by how likely the policy will take that action. Lastly, we need to estimate the true Q-function, which is achieved via a Monte Carlo sample of the observed rewards:

$$\nabla_\pi J(\pi) \propto \mathbb{E}_{s \sim d^\pi(s), a \sim \pi(\cdot|s)} \left[ (\nabla_\pi \log \pi(a|s)) \sum_{i=t}^{N} r_i \right] \tag{30}$$

The REINFORCE algorithm collects on-policy data and trains with $L_\pi = -\log \pi(a|s) \sum_{i=t}^{N} r_i$ as loss.

Vanilla REINFORCE policy gradients have high variance because they use Monte Carlo samples to estimate the Q-function and expectation. Revisiting the chess example, if the policy plays a perfect opening but makes a mistake in the middle game and loses, then all opening moves will be down-weighted as well. Discovering which actions contributed to the return is called the *credit assignment problem*. Credit assignment techniques, while not strictly necessary, often significantly simplify optimization. The objective $G$, here the return $\sum_{i=t}^{N} r_i$, is the feedback signal from the environment used to "reinforce" the actions. One way to reduce the variance of the gradient is to normalize the objective $G$ by subtracting a baseline $b(s)$, which can be any constant or function that depends on the state (Williams, 1992; Weaver & Tao, 2001):

$$G = \sum_{i=t}^{N} r_i - b(s) \tag{31}$$

Baselines do not bias the gradient in expectation, because the policy is a probability distribution (Weaver & Tao, 2001; Greensmith et al., 2001; 2004), but often reduce variance. An approximation of the value function $V$ is the most commonly used baseline, we will discuss it in more detail in Section 6.2.4. Conditioning the baseline additionally on the action has not been found to reduce variance in practice (Tucker et al., 2018).

## 6.2 Common Problems and Solutions

REINFORCE has a number of drawbacks, that the community tried to address. In the following, we will discuss these issues and popular extensions to REINFORCE.

### 6.2.1 Sample Efficiency

One of the drawbacks of REINFORCE is that the algorithm is on-policy and hence sample inefficient. A technique called *importance sampling* can be used to train policy gradient methods with off-policy data (Meuleau et al., 2001). Importance sampling makes the assumption that the policy that collected the data has a non-zero chance of picking any action and that the distribution is known. Using these assumptions, there is always a non-zero chance that the action that would have been sampled from the policy that you are currently training and the action that the data collecting policy $\pi_\beta$ sampled are the same. Importance sampling corrects the difference between the policies by scaling the loss according to the ratio between the respective probabilities.

$$L_\pi = -\frac{\pi(a|s)}{\pi_\beta(a|s)} \log \pi(a|s) \sum_{i=t}^{N} r_i \tag{32}$$

In the on-policy case where $\pi = \pi_\beta$ importance sampling simply multiplies by one, hence has no effect. Eq. (32) is usually simplified by reversing the log derivative trick from Eq. (29):

$$L_\pi = -\frac{\pi(a|s)}{\pi_\beta(a|s)} \sum_{i=t}^{N} r_i \tag{33}$$

Here, $\pi_\beta$ is constant wrt. the gradient computation. Importance sampling can increase sample efficiency since off-policy data can now be used. However, it has to be used with care. If the ratio between the policies for a particular state is small or large, importance sampling can lead to vanishing or exploding gradients. Importance sampling does not enable training with entirely off-policy or offline data, but empirically enables reuse of data samples for a couple of gradient steps. Computing the policy gradient with importance sampling is only an approximation to the off-policy policy gradient when the policy is a deep neural network (Degris et al., 2012). Additionally, the observed return $\sum_{i=t}^{N} r_i$ estimates $Q^{\pi_\beta}$ and not $Q^\pi$, when training with off-policy data. Algorithms that use importance sampling, like proximal policy optimization discussed in Section 6.3, use additional mechanisms to ensure that the data collecting policy $\pi_\beta$ and $\pi$ stay similar.

### 6.2.2 Reward Locality

The Monte Carlo return can lead to high variance gradients because future returns also reinforce current actions, even when future actions are responsible for the reward. A common assumption made is that actions

cannot change past rewards. This is a weak assumption that holds in most environments and is easy to verify. The sum of the objective $G$ starts at the current time step $t$:

$$G = \sum_{i=t}^{N} r_i \tag{34}$$

Another common assumption is that actions have a local effect. This means that the reward $r_i$ at time step $i$ is assumed to be most influenced by the action $a_i$ taken at that time step, or the actions shortly before. Mathematically this assumption can be incorporated by down-weighting future rewards exponentially using a *discount factor* $\gamma \in [0, 1]$ (Pardo et al., 2018) as already discussed before:

$$G = \sum_{i=t}^{N} \gamma^{i-t} r_i \tag{35}$$

Discount factors also limit the maximum time horizon that is optimized for, since the contribution of the reward will approach zero for larger $i$. Using discount factors distorts the objective and hence biases the solution. The advantage is that the model may learn faster since actions are only reinforced by the next few rewards. The best value of the discount factor depends on the environment and reward function. While tuning is required in practice, an initial value of 0.99 is often recommended (Andrychowicz et al., 2020). Again, the discount factor is a general idea that can be combined with any RL algorithm that addresses multistep problems. For clarity, we will set the discount factor to 1 in the following.

The assumption of reward locality does not hold in all environments. Consider for example again the game of chess where the reward is +1 for a win and -1 for a loss at the end of the game and 0 otherwise. Using a discount factor in such an environment does not ease learning, as all rewards except for the last one are zero. The only effect of a discount factor here would be that the objective prefers fast wins over wins in games with more steps. Like in off-policy learning, one way to address the problem of sparse rewards is to change the reward function such that it becomes more local and hence denser. This is called *reward shaping* (Ng et al., 1999). In chess, capturing of a piece could be positively rewarded while losing a piece could be negatively rewarded. The action of capturing a piece now is local with respect to the reward for capturing a piece. However, in chess not all captures are beneficial in the long term. In general, reward shaping biases the solution wrt. the original reward function, but can lead to better policies given a fixed amount of computation because learning can be faster. Like discount factors, reward shaping can be used in combination with any RL method.

### 6.2.3 Uncertain Futures

The return is typically an ambiguous objective, because the future is stochastic. Even the optimal action in a state may lead to a negative outcome later on. Consider again the example of chess at the first move. The network may lose even if it plays the optimal opening move by making a mistake later on. Additionally, the outcome of the game is highly dependent on the plays of the opponent. The Monte Carlo objective in Eq. (34) therefore has high variance. As a consequence, the same action in a state may receive very different gradients depending on the outcome of the episode. One way to mitigate this problem is to learn a second neural network, the value function $V^\pi(s)$, to directly predict the sum of rewards:

$$L_{V_\pi} = \left( \sum_{i=t}^{N} r_i - V^\pi(s) \right)^2 \tag{36}$$

The gradients of the value function suffer from the same uncertainty problem as before. To yield minimal training error, the value function tends to converge to the average return, interpolating between observed returns. Hence, the value function reduces the variance of the policy gradient by setting the objective to:

$$G = r_t + V^\pi(s') \tag{37}$$

This is a biased objective because the neural network estimating the value function can make mistakes. The value function and policy is trained in an iterative fashion, such that the estimate is particularly bad early

during training. However, using such a biased objective can pay off given a limited compute budget, because the policy gradient objective has reduced variance. In many environments, the immediate future is highly predictable, hence the bias of the objective can be reduced by using the next $n$ rewards instead:

$$G_n = \left( \sum_{i=t}^{t+n-1} r_i \right) + V^\pi(s_{t+n}) \tag{38}$$

This objective is called the *n-step* return where $n$ is a hyperparameter between [1, N] with $n = N$ recovering the original Monte Carlo objective in Eq. (34). All rewards after a terminal state are 0 which we have omitted for clarity. The hyperparameter $n$ trades bias against variance and typically has to be tuned per environment. As there might not be a single ideal value of $n$, a weighted average of all $n$-step objectives can also be used. The weight of each $n$-step return is typically decreased exponentially in $n$ using a hyperparameter $\lambda \in [0, 1]$:

$$G_\lambda = (1 - \lambda) \left( \sum_{n=1}^{N-t-1} \lambda^{n-1} G_n \right) + \lambda^{N-t-1} \sum_{i=t}^{N} r_i \tag{39}$$

$\lambda = 0$ recovers Eq. (37) (we define $0^0 = 1$), while $\lambda = 1$ recovers the standard Monte Carlo return. This *λ-return* objective is also called an *eligibility trace*. Both $n$-step returns and eligibility traces are not specific to policy gradient methods but can be used with any method that learns a value function. Using a value function to predict expected returns as targets or using discount factors are ways to deal with the uncertainty of returns. Some works try to learn the distribution of returns instead, although this idea is mostly used in the context of Q-learning (Bellemare et al., 2017; Dabney et al., 2018b;a).

### 6.2.4 Variable Episode Length

The magnitude of reinforcement of an action computed by the value function or return are dependent on the remaining length of the episode. Actions towards the end of an episode will receive a much smaller signal compared to actions at the beginning of the episode, in particular if the rewards are dense. A way to address this problem, is to use what is called the *advantage function* $A^\pi(s, a)$ as target (Baird, 1993). The advantage function describes the difference in expected return for each action $a$ in the current state $s$ when following $\pi$, compared to the expected return of the policy $\pi$ in state $s$. If any action has an advantage larger than 0, increasing the probability of the action in this state would improve the policy. Computing the policy gradient with the advantage function reinforces the current action only by the amount that it contributes to the return, as compared to the current policy, and hence disentangles the contribution of future actions. The advantage function itself is of course unknown, but we can approximate it by using the Q and V functions previously defined:

$$A^\pi(s, a) = Q^\pi(s, a) - V^\pi(s) \tag{40}$$

Both the $Q$ and value function $V$ could be estimated with Monte Carlo samples but two independent samples that start in the same state would be needed, otherwise the advantage is always 0. Usually, only one sample is available per state, so these functions are learned instead. Instead of estimating two functions, we can compute the advantage function using a single neural network for the value function by using the TD(0) idea to estimate the Q-function:

$$
\begin{aligned}
A^\pi(s, a) &= r_t + V^\pi(s') - V^\pi(s) \\
&= G_{\lambda=0} - V^\pi(s)
\end{aligned}
\tag{41}
$$

In this case, the $\max_{a'} Q(s', a')$ from Q-Learning is replaced by $V^\pi(s')$ since we are not trying to learn the Q-value of the optimal policy but rather estimate the Q-value of the current policy. Instead of using $TD(0)$, which is equivalent to $\lambda = 0$, advantage estimation can also be combined with any $\lambda$-return which is called *Generalized Advantage Estimation* $A^\pi_\lambda$ (Schulman et al., 2016; Peng et al., 2018):

$$A^\pi_\lambda = G_\lambda - V^\pi(s) \tag{42}$$

We can replace the Q-function in Eq. (29) with the advantage function because they only differ by the value function. Subtracting $V^\pi(s)$ is simply a particular choice of baseline $b(s)$ in Eq. (31) (Mnih et al., 2016).

Another motivation for the advantage function is that the value function is an almost optimal choice of baseline for reducing the variance of the policy gradient (Schulman et al., 2016).

In addition to varying lengths of episodes due to reaching terminal states, environments often also have a maximum number of environment steps after which no reward is obtained. It might not be possible to infer this time limit from observations, which makes value estimation hard. However, this can easily be resolved by including the remaining time in the state $s$ (Pardo et al., 2018).

## 6.3    Proximal Policy Optimization (PPO)

Various methods exist that build upon the basic policy gradient idea (Mnih et al., 2016; Espeholt et al., 2018; Petrenko et al., 2020; Cobbe et al., 2021). In this section, we cover a concrete example, proximal policy optimization (PPO) (Schulman et al., 2017), which is a widely used RL algorithm that combines many extensions to the classic REINFORCE algorithm that we discussed in Section 6.1. PPO has been used successfully to master complex games (Berner et al., 2019), learn autonomous driving planners (Zhang et al., 2021), control drones (Kaufmann et al., 2023) or tune large language models (Ouyang et al., 2022).

The core learning mechanism of PPO is the on-policy policy gradient objective from Section 6.1. The objective is the advantage function, learned via generalized advantage estimation, as introduced in Section 6.2.4. The advantage function and the value function are both estimated using $\lambda$-returns, as discussed in Section 6.2.3. To increase sample efficiency, training is done via $M$ different actors collecting data in parallel. $T$ time steps are collected by each actor and stored in a buffer $B$, creating a small dataset. If an episode does not end before $T$ steps, its remaining return will be estimated with a value function. If an episode ends before $T$ steps, the end is marked and a new episode will begin immediately. This ensures that always $T \times M$ samples are collected, enabling efficient vectorization of computation. The buffer $B$ is treated as a small dataset and trained with for $K$ epochs using mini-batches. After training for $K$ epochs, the data collecting policy $\pi_\beta$ is updated, the data is deleted, and the training loop begins its next iteration. Environments are not reset at every iteration, and rather resumed where the previous iteration stopped. This ensures that environments with time horizons $N > T$ will finish. Training for multiple epochs increases sample efficiency but introduces a divergence between the data collecting policy $\pi_\beta$ and the training policy $\pi$, the data is off-policy. This difference is mitigated by using importance sampling, as discussed in Section 6.2.1. To keep the data collecting policy $\pi_\beta$ and the training policy $\pi$ close, the number of epochs $K$ is typically small. Additionally, PPO uses the following clipping mechanism for its objective function, called the *PPO-clip* objective:

$$L_\pi = -\min\left(\frac{\pi(a|s)}{\pi_\beta(a|s)}A_\lambda^\pi, \text{clip}\left(\frac{\pi(a|s)}{\pi_\beta(a|s)}, 1.0 - \psi, 1.0 + \psi\right)A_\lambda^\pi\right) \tag{43}$$

The clipping threshold $\psi$ is a hyperparameter that is set to 0.2 by default. PPO-clip is illustrated in Fig. 5.

The x-axis represents the probability of $\pi$ and the y-axis represents the loss. The different columns illustrate different values of $\pi_\beta$. The loss explodes once $\pi_\beta$ becomes small due to the importance sampling correction. The top row illustrates the loss for positive advantages, the bottom row illustrates it for negative once. The goal of PPO-Clip is to prevent the training policy $\pi$ to diverge too far from the data collecting policy $\pi_\beta$. When the advantage is positive, the probability of the action will be increased. The effect of PPO-Clip is that it sets the gradient to zero if $\pi(a|s) > (1+\psi)\pi_\beta(a|s)$. Analogously, if the advantage is negative, the probability of the action will be decreased and PPO clip will set the loss to a constant value if $\pi(a|s) < (1 - \psi)\pi_\beta(a|s)$ (Achiam, 2018b). If $\pi(a|s)$ changes from $\pi_\beta(a|s)$ in the "wrong" direction, e.g., making an action with a positive advantage less likely due to function approximation, then the loss will not be clipped by PPO-clip. For example, if $\pi_\beta(a|s) = 0.5$ and the advantage is negative, but the policy changed to $\pi(a|s) = 0.9$ due to a function approximation error, then the loss is not clipped, and the policy can be corrected even though $\pi$ was far away from $\pi_\beta$.

Besides the clipping objective, future returns are smoothed by learning a value function:

$$L_V = (G_\lambda - V^\pi(s_t))^2 \tag{44}$$

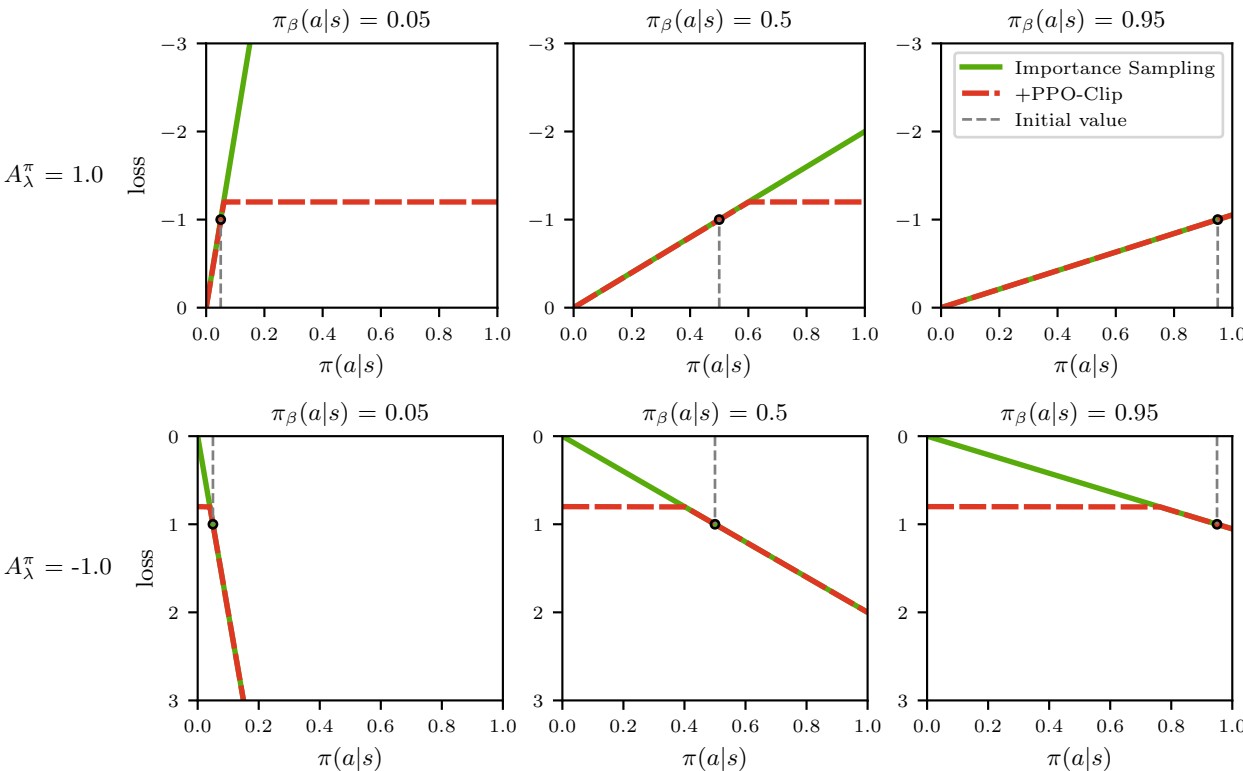

Figure 5: **PPO-Clip objective.** Top row illustrates positive, bottom row negative advantages. The columns illustrate different probabilities of the data collecting policy $\pi_\beta$. Optimization moves upwards. PPO-Clip clips the objective if $\pi$ moved too far upwards compared to $\pi_\beta$. Here, we use a clipping threshold $\psi = 0.2$

PPO further encourages exploration by increasing the entropy of $\pi(a|s)$ (Williams & Peng, 1991):

$$L_{\mathcal{H}} = \sum_{a \in A} \pi(a|s) \log \pi(a|s) \tag{45}$$

The value function and the policy function can be heads of a single neural network that share a common backbone. With shared backbones, the following joint loss is minimized during training:

$$L_{PPO} = L_\pi + c_1 L_V + c_2 L_{\mathcal{H}} \tag{46}$$

where $c_1$ and $c_2$ are tune-able hyperparameters. PPO also uses discount factors and is often trained with shaped rewards, as discussed in Section 6.2.2. The PPO algorithm is illustrated in Algorithm 2.

# 7 Discussion

Training deep neural networks has become the standard recipe to solve machine learning tasks. Reinforcement learning enables optimization of deep neural networks for non-differentiable objectives, giving users a lot of freedom to closely align training objectives with the desired outcome. We have introduced the two core ideas of RL that enable optimization of non-differentiable metrics, value learning and policy gradients. These techniques are commonly used in natural language processing (Bahdanau et al., 2017) and have seen early successes in computer vision (Huang et al., 2021; Pinto et al., 2023) but due to their generality are relevant for any machine learning task.

RL techniques were initially designed for sequential decision making tasks. Most modern RL algorithms in this setting are motivated by either the $Q$-learning theorem (Watkins & Dayan, 1992) or the policy gradient theorem (Sutton et al., 1999; Marbach & Tsitsiklis, 2001).

---

**Algorithm 2** PPO

---

**for** iterations **do** ▷ Training loop
    Create empty temporary buffer $B$
    **for** actors=1,2,...,$M$ **do** ▷ Run in parallel
        Collect $T$ time steps of data with $\pi_\beta$ stored in temporary buffer $B$
    **end for**
    Compute $G_\lambda = \left( (1-\lambda) \sum_{n=1}^{N-t-1} \lambda^{n-1} \left( \sum_{i=t}^{n+t-1} \gamma^{i-t} r_i + \gamma^n V^\pi \left( s_{t+n} \right) \right) + \lambda^{N-t-1} \sum_{i=t}^{N} \gamma^{i-t} r_i \right)$
    Compute $A_\lambda^\pi = G_\lambda - V^\pi(s_t)$
    **for** epochs=1,2,...,K **do** ▷ Train K times per sample
        **for all** mini-batches $\in B$ **do** ▷ Sample mini-batches from collected data
            Compute $L_\pi = -\min\left( \frac{\pi(s)}{\pi_\beta(s)} A_\lambda^\pi, \text{clip}\left( \frac{\pi(s)}{\pi_\beta(s)}, 1.0-\psi, 1.0+\psi \right) A_\lambda^\pi \right)$
            Compute $L_\mathcal{H} = \sum_{a \in A} \pi(a|s) \log \pi(a|s)$
            Compute $L_V = (G_\lambda - V_\pi(s_t))^2$
            Train network to minimize $L_{PPO} = L_\pi + c_1 L_V + c_2 L_\mathcal{H}$
        **end for**
    **end for**
    $\pi_\beta \leftarrow \pi$
**end for**

---

The Q-learning theorem allows us to find the optimal policy even with off-policy data, but unfortunately requires the Q-function to be represented by a table and so does not apply to deep neural networks. Fortunately, it turns out that with the right empirical techniques, it is possible to train deep Q-networks with Q-learning as well (Mnih et al., 2015; Fan et al., 2020). We have discussed the important problems that arise when training deep Q-networks and presented Soft Actor-Critic (Haarnoja et al., 2018a;b), a popular algorithm that incorporates many of the solutions.

The policy gradient theorem requires the data to be collected on-policy, except for 1-step environments, but crucially applies to deep neural networks. On-policy learning is sample inefficient because samples can not be reused and gradients have high variance due to the use of Monte Carlo sampling. Many techniques have been proposed to reduce the variance of the policy gradient and improve sample efficiency. We presented Proximal Policy Optimization (Schulman et al., 2017), a popular algorithm that combines these techniques into one method.

In general, there is no clear guide on what RL algorithm to use for a specific problem. PPO may be a good first choice since the algorithm has seen success on a wide range of different problems (Berner et al., 2019; Zhang et al., 2021; Ouyang et al., 2022; Kaufmann et al., 2023). In specific domains, like continuous control with low dimensional state spaces, Soft Actor-Critic has yielded better performance (Haarnoja et al., 2018a).

**Limitations:** This article focused on understanding the core ideas of RL. As such, we have focused on intuitive explanations and readable equations instead of theoretical rigor and general equations. Besides the core ideas, training a successful reinforcement learning policy for sequential decision making tasks additionally involves a lot of engineering and implementation tricks, like gradient clipping, in particular for PPO (Henderson et al., 2018; Andrychowicz et al., 2020; Engstrom et al., 2020; Huang et al., 2022a). Additionally, the performance of these algorithms can be quite sensitive to the initial random seed. We encourage RL researchers to use appropriate statistical methods to evaluate the performance of their algorithms (Agarwal et al., 2021). We also encourage practitioners interested in applying RL to their problem to start with open source implementations (Dhariwal et al., 2017; Castro et al., 2018; Raffin et al., 2021; Huang et al., 2022b) to avoid pitfalls in reproducing existing algorithms. Reinforcement learning techniques require a lot of data even with off-policy methods, so most successful applications of RL involved a simulator. Reducing the number of required samples (Kaiser et al., 2020) or training policies for sequential problems with entirely offline data is currently an active area of research (Gürtler et al., 2023; Prudencio et al., 2023).

The field of RL is over 40 years old and has developed a wide range of methods. We have only covered the most important ideas of the field. Some interesting additional topics, like reinforcement learning from human feedback, that are more niche, are discussed in the appendix. Sutton & Barto (2018) is a great resource for readers interested in methods that use tables or linear function approximation. As additional resources, the RL community provides free lectures (Silver, 2015; Levine, 2023) and websites (Achiam, 2018a).

## Broader Impact Statement

Reinforcement learning can be used to directly optimize models for complex objectives. This may lead to undesirable behavior if the mathematical implementation of the objective is just a proxy for the original goal, which can be the case for more complex metrics. It is therefore important to not just rely on quantitative metrics for evaluation, but also qualitatively test if the trained model has learned the desired behavior.

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

# Appendix

Here, we briefly introduce some additional topics from the field of reinforcement learning that are important, but more niche than the ideas covered in the main paper.

## A   Upside Down RL

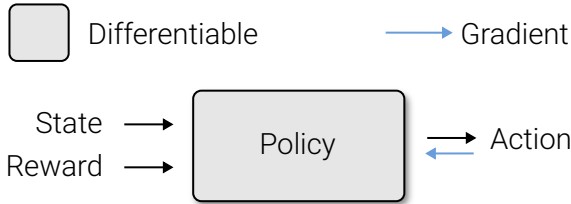

Figure 6: **Upside down RL** conditions on the reward.

Upside Down RL (Schmidhuber, 2019; Srivastava et al., 2019; Kumar et al., 2019) is a different but simple concept to bridge the non-differentiable gap between the action and a reward. The idea is to use the reward as conditioning input of the policy:

$$L_\pi := \|a - \pi(s, r)\|_2^2 \qquad (47)$$

Additionally, the number of steps in an episode can be added to the input. The policy network is then simply trained with supervised learning, predicting the action that achieved the given reward in this state. In sequential problems, the return can be used for conditioning. Upside Down RL is illustrated in Fig. 6. During inference, the reward is simply set to the maximum reward to obtain the best action. Upside down RL is a relatively new idea and still part of ongoing research. It has seen most success when combined with transformers in offline RL settings (Chen et al., 2021).

## B   Model-based Reinforcement Learning

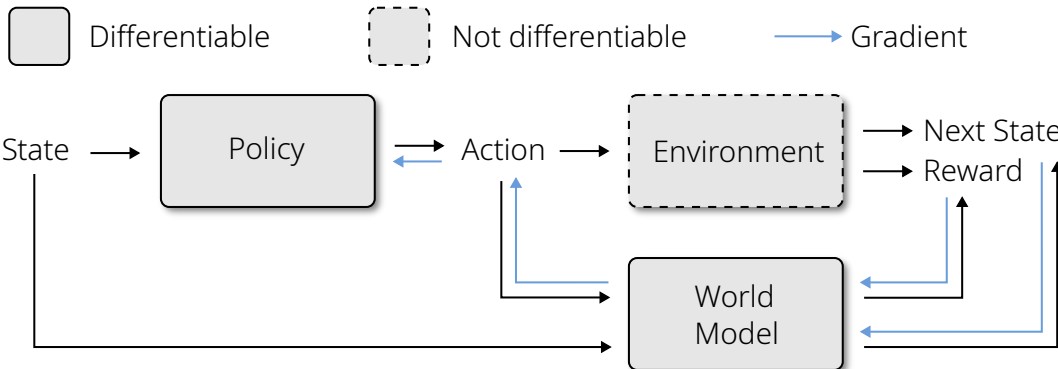

Figure 7: **Model-Based RL** learns the environment self-supervised.

In model-based RL (Ha & Schmidhuber, 2018; Hafner et al., 2020; 2021; 2023), the non-differentiable environment gap is bridged by learning the environment dynamics explicitly via self-supervised learning. A differentiable model, called the *world model* is optimized to predict the next state and reward, given the current state and action. Compared to model free methods, much richer labels are available because the next state is usually high dimensional. The world model can then for example be used to maximize the return inside the world model directly because it is differentiable. This is illustrated in Fig. 7. Backpropagating

through long time horizons can be computationally expensive if the world model has many parameters. A world model can also be used as a learned simulator, which offers a way to generate large amount of samples when environment interaction with the real system is limited. A disadvantage of model-based RL is that the policy can and will exploit inaccuracies in the world model. For example, if the world model incorrectly attributes a lot of reward to an action, the policy trained inside the world model will pick that action even when this action is suboptimal in the real environment. Inaccuracies in the predicted observations can also be problematic if small details in the input are relevant for the downstream task. The world model might not learn small details because they have a low impact on the loss for predicting the next state. Despite the downsides, model-based RL can be useful in settings where the number of available interactions with the real environment is limited.

## C  Reinforcement Learning from Human Feedback

Reinforcement Learning from Human Feedback (RLHF) (Christiano et al., 2017; Stiennon et al., 2020; Bai et al., 2022; Ouyang et al., 2022) describes the idea of using rankings from human annotators as the target objective to optimize or fine-tune a model. The optimization uses a combination of the standard reinforcement learning ideas discussed in the main text. RLHF is primarily used to optimize generative models in particular large language models, thus we will focus our discussion on the particular considerations of that task. RLHF has been an integral technique used to turn large language models into useful products like ChatGPT. Similar ideas have also been applied to models that generate images (Black et al., 2023; Fan et al., 2023; Wallace et al., 2023).

Large language models (LLM) (Brown et al., 2020) are trained to predict the probability of the next word, or parts of words called tokens, given prior words in a sentence. This is a self-supervised objective which enables training on internet scale datasets. At inference, these models can be used to generate text by iteratively sampling a word from the predicted distribution. This generates plausible sounding text given an initial text, called the *prompt*. Generating plausible continuations of text can be useful because, for example, the correct answer to a question contains some of the most likely words. However, the correct answer is not the only plausible continuation of the text. The large scale datasets from the internet that LLMs are trained with also contain lies, offensive speech, manipulative or simply unhelpful text. LLMs trained in a self-supervised way on this data may therefore also generate such responses, and are therefore not safe to deploy into products for end users. One remedy to this problem is *supervised fine-tuning* (SFT) where a labeled dataset with prompts and target texts from a human annotator is collected and trained with. SFT has limited effectiveness because creating large labelled datasets with demonstrations is expensive. Additionally, individual human annotators have limited skill sets, for example they don't know the correct answer to every question for which a correct answer is known and available on the internet. A more scalable approach is to collect a dataset where the pre-trained model generates multiple responses to a given prompt, with its internet scale knowledge base. The human annotators are then tasked to rank these predictions from best to worst. This approach is more scalable because it is easier for humans to verify the correctness of an answer rather than coming up with the correct answer from scratch. However, maximizing human rankings is not a differentiable objective, which is where reinforcement learning comes to the rescue.

In the version of RLHF proposed by Ouyang et al. (2022) a reward model is first learned from a dataset containing human rankings. The reward model predicts the ranking given a prompt and an answer sampled from the model. This is a form of value learning, where the reward model can be thought of as a Q-function. Learning this Q-function is very hard because for example the Q-function needs to know which of the presented answers is correct, to predict which one the human would prefer. The task is made possible by using a pre-trained LLM as the architecture for the Q-function, with minimal modification to be able to predict rankings. LLMs are probabilistic models, so stochastic policy gradients are used to tune them. In particular, Ouyang et al. (2022) uses the PPO algorithm discussed in Section 6.3.

RLHF combined with supervised fine-tuning has been found effective enough to deploy LLM chatbots on a large scale. The goal of RLHF is, given a learned distribution, to "unlearn" the parts of the distribution that are considered bad behavior. Current RLHF is far from perfect and an active field of research (Rafailov et al., 2023; Wu et al., 2023). Models do not forget all harmful parts of the distribution and also tend to

unlearn useful predictions. This is mitigated by mixing RLHF gradients with gradients from the original self-supervised pre-training Ouyang et al. (2022). It is worth noting that with RLHF a generative model is unlikely to learn new behavior as it only reinforces predictions that the generative model has already been capable of generating.

## D    Planning

To find the optimal action $a^\star$ for each state $s$ we have primarily considered *policies*, the approach of learning a function $\pi$ that maps states to actions. There is another approach called planning, which describes *algorithms* that given a model of the environment find the optimal action or improve the actions of a policy.

### D.1    Tree Search

A powerful class of algorithms are *search* algorithms, out of which *tree search* is arguably the simplest. Tree search requires a world model that given a state and action can predict the next state and reward. This can be a learned world model, but it does not have to be differentiable, a classic simulator also works. Given a state $s$, the tree search algorithm computes the next state and stores the reward, for every possible action. In this naive version, the action space has to be discrete. The process of simulating the next time step for every possible action is then repeated for all possible next states from the previous iteration until all branches of this tree have finished in a terminal state. The observed rewards are then used to choose the action from the first iteration based on some criterion, such as highest average return. If the environment has deterministic state transitions, the action space is discrete and the world model perfect, then this algorithm will find the optimal action $a^\star$. This process will then be repeated for the next state, potentially reusing simulations from prior steps. The difficulty of tree search is that the algorithm will exploit any inaccuracies in the world model, and most importantly it is too slow to run for complex environments. Exhaustively simulating all potential futures is not possible in most cases. In the following section, we will describe a more practical class of search algorithms that use the idea of Monte-Carlo sampling (Metropolis & Ulam, 1949) to efficiently choose which futures to evaluate.

### D.2    Monte-Carlo Tree Search

The core idea of *Monte-Carlo tree search* (MCTS) (Coulom, 2006) is to only explore a part of the full tree by using heuristics and random actions to choose which states and actions to evaluate.

MTCS starts by creating a tree with the current state as the root node and iteratively repeats the following 4 steps until a certain time limit or resource constraint is met.

1. **Selection.** A *tree policy* is used to select a state which still has at least one unexplored action.

2. **Expansion.** An unexplored action in that state is chosen, expanding the tree.

3. **Simulation.** The next action is chosen iteratively by a probabilistic policy until the episode ends.

4. **Backup.** The nodes, up until the node starting the simulation, are updated with the return.

Fig. 8 illustrates these 4 steps. The probabilistic policy, also called the *default policy*, from step 3 can be any probabilistic policy but should be fast to evaluate for the whole process to be efficient, so simple linear layers (Silver et al., 2016) or just a uniform distribution are used in practice. MCTS may start with an empty tree if the current state is novel. If the current state already was a node in the tree from the previous iteration, then that node is used as the root node of the new tree and its children are retained. Modern implementations of MCTS combine the idea with policies and value functions trained with reinforcement learning (Silver et al., 2016; 2017; 2018) as well as learned world models (Schrittwieser et al., 2020).

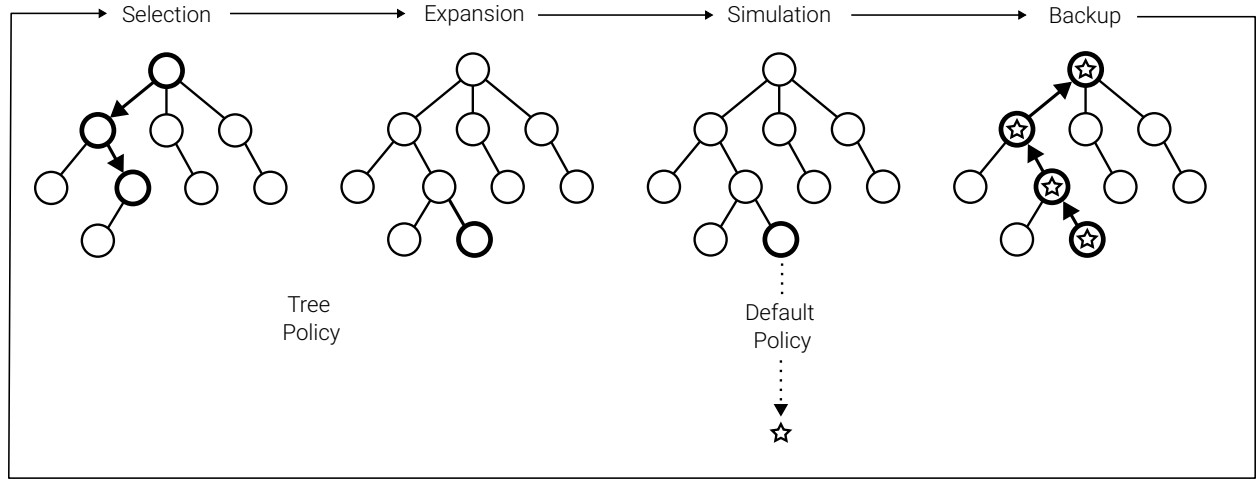

Figure 8: **Monte Carlo Tree Search.**

## E   Related Work

The most widely cited introduction to reinforcement learning is Sutton & Barto (2018) which is a 526-page-long textbook. It puts a strong focus on theoretical foundations and methods using tabular representations or linear function approximation. For such problems, much stronger theoretical guarantees can be obtained than for the non-linear function approximation problems that we considered in this work. The textbook also discusses applications of RL in psychology and neuroscience. As such, Sutton & Barto (2018) is complementary to this work, and we recommend it for readers that want to deeply familiarize themselves with the field. A lot of introductions (Mousavi et al., 2016; Li, 2017; 2018), books (Szepesvári, 2010; François-Lavet et al., 2018; Sutton & Barto, 2018), surveys (Kaelbling et al., 1996; Arulkumaran et al., 2017; Wang et al., 2020a; Lazaridis et al., 2020; Joshi et al., 2021; Wang et al., 2022) and tutorials (Harmon & Harmon, 1996; Li & Pyeatt, 2004; Gosavi, 2009; Levine et al., 2020; Vidyasagar, 2023) have been written over the years on the topic of RL. This work adds to the literature by introducing RL from the alternative angle of optimizing non-differentiable objectives. We introduce the reader to the most important ideas in the field and show the close connection between supervised learning and RL.

## F   Bellman Equations

Throughout this tutorial, we have covered some special cases to keep the equations simple. In this section, we will briefly discuss a more general form of the value learning formalism and introduce the Bellman equations. Our setting is a *Markov Decision Process* (MDP) $(S, A, P, R)$, where $S$ is the set of all states, $A$ the set of all actions, $P(s', s, a) = Pr(s_{t+1} = s' | s_t = s, a_t = a)$ an environment transition kernel, describing the probability of the next state given the current state and action and $R(r|s, a) = Pr(r_t = r | s_t = s, a_t = a), r \in \mathfrak{R}$ is the reward function and $\mathfrak{R} \subset \mathbb{R}$ the set of all rewards. The goal is to find the optimal policy $\pi^\star(s) \to a$, that for every state $s \in S$ maps to an action $a \in A$ that is optimal wrt. the expected discounted return $G$:

$$G_t = \sum_{k=t}^{T} \gamma^{k-t} r_k \tag{48}$$

Here, $\gamma \in [0, 1]$ is a discount factor and $r_k \in \mathfrak{R}$ is the observed reward at time step $k$. If the environment has infinite length $T = \infty$ then $\gamma$ must be $\in [0, 1)$ to avoid infinite sums. Policies may be probabilistic, in which case they are denoted by $\pi(a|s)$. To measure the expected return of a policy $\pi$ a so-called value function $V^\pi(s)$ is often used:

$$V^\pi(s) = \mathbb{E}_\pi [G_t | s_t = s] \tag{49}$$

Here, $\mathbb{E}_\pi$ denoted the expected value when the agent starts in state $s$ and follows the policy $\pi$ at every time step. Methods that try to extract the optimal policy $\pi^\star$ from a value function typically condition it on the first action, which is called the Q-function or action-value function:

$$Q^\pi(s, a) = \mathbb{E}_\pi\left[G_t | s_t = s, a_t = a\right] \tag{50}$$

An important property of value functions is that they can be expressed recursively in the following form:

$$
\begin{aligned}
V^\pi(s) &= \mathbb{E}_\pi\left[G_t | s_t = s\right] \\
&= \mathbb{E}_\pi\left[r_t + \gamma G_{t+1} | s_t = s\right] \\
&= \sum_{a \in A} \pi(a|s) \sum_{s' \in S} \sum_{r \in \mathfrak{R}} Pr(s_{t+1} = s', r_t = r | s_t = s, a_t = a) \left(r + \gamma \mathbb{E}_\pi\left[G_{t+1} | s_{t+1} = s'\right]\right) \\
&= \sum_{a \in A} \pi(a|s) \sum_{s' \in S} \sum_{r \in \mathfrak{R}} Pr(s_{t+1} = s', r_t = r | s_t = s, a_t = a) \left(r + \gamma V^\pi(s')\right)
\end{aligned}
\tag{51}
$$

Eq. (51) is called the Bellman equation for $V^\pi$. It is the motivation for a number of ways to learn or approximate $V^\pi$.

The value function of the optimal policy can be expressed in terms of the Q-function of the optimal policy, by taking an optimal action in the current time step and following the optimal policy afterward:

$$
\begin{aligned}
V^{\pi^\star}(s) &= \max_{a \in A} Q^{\pi^\star}(s, a) \\
&= \max_{a \in A} \mathbb{E}_{\pi^\star}\left[G_t | s_t = s, a_t = a\right] \\
&= \max_{a \in A} \mathbb{E}_{\pi^\star}\left[r_t + \gamma G_{t+1} | s_t = s, a_t = a\right] \\
&= \max_{a \in A} \mathbb{E}\left[r_t + \gamma V^{\pi^\star}(s_{t+1}) | s_t = s, a_t = a\right] \\
&= \max_{a \in A} \sum_{s' \in S} \sum_{r \in \mathfrak{R}} Pr(s_{t+1} = s', r_t = r | s_t = s, a_t = a) \left(r + \gamma V^{\pi^\star}(s')\right)
\end{aligned}
\tag{52}
$$

This is called the Bellman optimality equation of $V^{\pi^\star}$. Similar, the Bellman optimality equation of $Q^{\pi^\star}$ is:

$$
\begin{aligned}
Q^{\pi^\star}(s, a) &= \mathbb{E}\left[r_t + \gamma V^{\pi^\star}(s') | s_t = s, a_t = a\right] \\
&= \mathbb{E}\left[r_t + \gamma \max_{a' \in A} Q^{\pi^\star}(s', a') | s_t = s, a_t = a\right] \\
&= \sum_{s' \in S} \sum_{r \in \mathfrak{R}} Pr(s_{t+1} = s', r_t = r | s_t = s, a_t = a) \left(r + \gamma \max_{a' \in A} Q^{\pi^\star}(s', a')\right)
\end{aligned}
\tag{53}
$$

If $Q^{\pi^\star}$ is known, it simplifies finding the optimal policy $\pi^\star$ to:

$$\pi^\star(s) = \operatorname*{argmax}_{a \in A} Q^{\pi^\star}(s, a) \tag{54}$$

Q-learning and other decision-making methods can be viewed as approximately solving the Bellman optimality equation to find $Q^{\pi^\star}$ (Sutton & Barto, 2018).

