# OpenReview forum: "An Invitation to Deep Reinforcement Learning"
_TMLR — Rejected by TMLR_

### Review · Reviewer_FkiF · 2024-01-12

**Summary Of Contributions:**

The paper presents an introduction to deep RL aimed at ML researchers
already familiar with supervised learning.
The paper covers the basics of value and policy learning
with on- and off-policy methods and connects back to the
idea of using policy gradients for image classification.

**Audience:**

No

**Claims And Evidence:**

No

**Requested Changes:**

1. I am marking this paper as not supporting claims as S3.1.3 and
  S3.2.1 discuss the use of policy learning and Q learning for
  image classification. Unless there is some variation on the standard
  classification setting, I strongly do not believe policy or Q
  learning methods will improve ResNet/image classifier training,
  but the table in S3.1.3 suggests that this may be a possibility.
  I request for this to be clarified, discussed, and/or for the
  table to be removed.
2. In many places (Fig 1 (sec 3.1), eq (1), eq (2)), the Q function is presented to
   be the reward, but this is not true in general.
   The Q function is the expected (and possibly discounted) future sum of rewards.
   Q learning is /not/ done as in eq (2) by just matching the reward values.
   Simplifying it to the reward is an over-simplification and
   misleading. These instances should be corrected.

**Strengths And Weaknesses:**

While the nascent field of deep RL benefits from introductory
approaches targeting the concepts at researchers with different
backgrounds, I find it difficult to evaluate this TMLR submission.
While I do not see any strong weaknesses to the presentation,
I also do not see any significant new content or presentation that
a reader would take out of it that is not in the existing literature.
For example, the paper mostly builds up to discussing the contents
around PPO and SAC, but I do not see a strong advantage to readers
learning about these methods via this new paper instead of reading
the original and other existing papers around PPO and SAC or
existing RL books or courses such as at [Berkeley](https://rail.eecs.berkeley.edu/deeprlcourse/),
or [CMU](https://cmudeeprl.github.io/703website_f23/).
I acknowledge that I am not the target audience of this paper as
I am already familiar with the RL foundations and methods, so
may not have been able to appreciate it as much as a newcomer to
the field would.

---

> ### Author Response · Authors · 2024-01-17
>
> Thank you very much for your feedback and suggestions. We respond to each of your concerns one by one below:
>
> > "While I do not see any strong weaknesses to the presentation, I also do not see any significant new content or presentation that a reader would take out of it that is not in the existing literature. For example, the paper mostly builds up to discussing the contents around PPO and SAC, but I do not see a strong advantage to readers learning about these methods via this new paper instead of reading the original and other existing papers around PPO and SAC or existing RL books or courses such as at Berkeley, or CMU.”
>
> Lecture courses serve a different purpose than tutorials, as they require a different order of time commitment. Taking a standard 6 credit lecture on the subject of RL is estimated to be a 150 - 180 hour time commitment.
> Reading this tutorial takes a couple of hours, so is 1–2 orders of magnitude more time efficient. We necessarily cover less content than a lecture course. The advantage of this tutorial is that the reader, given a minimal time commitment, will be exposed to only the essential ideas and knows enough to apply and properly understand popular RL methods.
>
> As we mention in the introduction (page 2), we would not recommend newcomers to the field of RL to start by reading the original papers, as they often focus on rigorous theoretical expositions over didactic clarity because they are written with an expert audience in mind. This means that the reader typically needs to be familiar with the relevant prior work to understand the method that is presented.
> For example, the PPO paper assumes the reader is familiar with policy gradients and why they work  (Williams, 1992; Sutton et al., 1999), why policy gradients require on-policy data and how importance sampling can be used to estimate the off-policy policy gradient (Degris
> et al., 2012), why this is still problematic and motivates clipping, what the advantage function is (Baird, 1993), how it reduces variance and how it is generalized using eligibility traces (Schulman et al. 2016) and why parallel data collection is important (Mnih et al. 2016).
> In contrast, our tutorial is self-contained and does not require prior knowledge from the field of RL, enabling the reader to know not just what these methods do but also understand why they are built this way.
>
> Many tutorial introductions of similar length (Harmon and Harmon 1996; Li and Pyeatt 2004; Gosavi 2009) are quite outdated and do not cover modern deep RL. Levine et al. (2020) focuses on offline RL, whereas we focus on more mature off-policy and on-policy RL techniques. As such, it is complementary to our work. Vidyasagar (2023) focuses on reinforcement learning theory and does not cover actor-critic methods or modern RL algorithms, whereas our focus lies on modern deep RL.
>
> RL Books, like lectures, typically require a larger time commitment to read. Taking the standard textbook Sutton and Barto (2018) as an example, which has 526 pages and hence is an order of magnitude longer than this tutorial. It puts a strong focus on theoretical foundations and methods using tabular representations or linear function approximation. We are able to introduce RL in a much more concise form by introducing it from the alternative angle of generalizing supervised learning to non-differentiable objective functions, and multiple time-steps  (see also appendix E, page 38).
>
> *To summarize*:
> Our tutorial adds to the literature by providing readers with a self-contained introduction to deep RL that is more concise than other material yet still covers all the important concepts for the reader to understand and apply modern deep RL algorithms. We achieve this by introducing deep RL through the lens of generalizing supervised learning to non-differentiable objectives and hope this perspective will facilitate adoption of deep RL in other fields where the connection to classical RL might be less obvious.

---

> ### Author Response · Authors · 2024-01-17
>
> References:
>
> Leemon C Baird. Advantage updating. Technical report, Technical report wl-tr-93-1146, Wright Patterson
> AFB OH, 1993.
>
> Thomas Degris, Martha White, and Richard S. Sutton. Off-policy actor-critic. arXiv.org, 1205.4839, 2012.
>
> Abhijit Gosavi. Reinforcement learning: A tutorial survey and recent advances. INFORMS J. Comput., 2009.
>
> Evan Greensmith, Peter L. Bartlett, and Jonathan Baxter. Variance reduction techniques for gradient estimates in reinforcement learning. Journal of Machine Learning Research (JMLR), 2004.
>
> Mance E Harmon and Stephanie S Harmon. Reinforcement learning: A tutorial. WL/AAFC, WPAFB Ohio, 1996.
>
> Sergey Levine, Aviral Kumar, George Tucker, and Justin Fu. Offline reinforcement learning: Tutorial, review, and perspectives on open problems. arXiv.org, 2020.
>
> Chengcheng Li and Larry D. Pyeatt. A short tutorial on reinforcement learning. In Intelligent Information Processing II, 2004.
>
> Volodymyr Mnih, Adrià Puigdomènech Badia, Mehdi Mirza, Alex Graves, Timothy P. Lillicrap, Tim Harley, David Silver, and Koray Kavukcuoglu. Asynchronous methods for deep reinforcement learning. In Proc. of the International Conf. on Machine learning (ICML), 2016.
>
> John Schulman, Philipp Moritz, Sergey Levine, Michael I. Jordan, and Pieter Abbeel. High-dimensional continuous control using generalized advantage estimation. In Proc. of the International Conf. on Learning Representations (ICLR), 2016.
>
> Richard S. Sutton, David A. McAllester, Satinder Singh, and Yishay Mansour. Policy gradient methods for reinforcement learning with function approximation. In Advances in Neural Information Processing Systems (NIPS), 1999.
>
> Richard S. Sutton and Andrew G. Barto. Reinforcement learning: An introduction. MIT press, 2018.
>
> Mathukumalli Vidyasagar. A tutorial introduction to reinforcement learning. arXiv.org, 2304.00803, 2023.
>
> Ronald J. Williams. Simple statistical gradient-following algorithms for connectionist reinforcement learning. Machine Learning, 8:229–256, 1992

---

> ### Author Response · Authors · 2024-01-17
>
> > "I am marking this paper as not supporting claims as S3.1.3 and S3.2.1 discuss the use of policy learning and Q learning for image classification. Unless there is some variation on the standard classification setting, I strongly do not believe policy or Q learning methods will improve ResNet/image classifier training, but the table in S3.1.3 suggests that this may be a possibility. I request for this to be clarified, discussed, and/or for the table to be removed."
>
> We are unsure how this confusion arises. We do not claim that policy gradients or Q-learning improve standard supervised classification, but instead emphasize the connection to supervised learning. We believe that viewing RL as a generalization of supervised learning helps to better understand their connection and makes RL techniques accessible to a larger readership. Section 3.2.1 argues that minimizing Cross-Entropy already maximizes accuracy, so there is no conceptual advantage to using vanilla policy gradients or Q-learning instead.
>
> The purpose of the table is to show that training classification models with a regression loss (MSE) works on a simple classification problem (Cifar-10). While MSE has been used in the past to train classification networks, it is uncommon today (section 3.1.3, page 8). So we think it is helpful for the reader to have a small experiment that shows that training a classification network with a regression loss is a sensible thing to do.
> We do not want to imply that the MSE loss is better than Cross-Entropy. MSE happens to have a slightly higher accuracy in this experiment, but +0.3 is not a significant difference on CiFAR-10 (section 3.1.3, page 8, “[...] Q-learning achieves similar accuracy to the cross-entropy (CE) loss.”).

---

> > ### Comment · Reviewer_FkiF · 2024-02-19
> >
> > >  While MSE has been used in the past to train classification networks, it is uncommon today (section 3.1.3, page 8). So we think it is helpful for the reader to have a small experiment that shows that training a classification network with a regression loss is a sensible thing to do.
> >
> > My comment is that I don't think it's sensible to train a classification network with Q learning, and it seems reasonable why it's not commonly used today. I still think it's distracting and potentially misleading to keep this section and experiment in the paper
> >
> > > We are unsure how this confusion arises. We do not claim that policy gradients or Q-learning improve standard supervised classification
> >
> > To me, the table in S3.1.3 showing that Q-learning improves upon standard supervised classification still strongly claims/implies that this is a research direction worth investigating, or a connection worth learning

---

> ### Author Response · Authors · 2024-01-17
>
> > "In many places (Fig 1 (sec 3.1), eq (1), eq (2)), the Q function is presented to be the reward, but this is not true in general. The Q function is the expected (and possibly discounted) future sum of rewards. Q learning is /not/ done as in eq (2) by just matching the reward values. Simplifying it to the reward is an over-simplification and misleading. These instances should be corrected."
>
> $Q(s,a) = \mathbb{E}[\sum^{T}_{i=t} \gamma^{i-t} r_i]$
>
> The Q-function simplifies to predicting the (expected) reward in these equations because we specifically only consider environments of length 1 in this section (single-step RL).
> Therefore, there is no future reward and the discount factor of the first reward is always 1.
>
> This is indicated in Figure 1 via the vertical bar on the left (“Single Step Problems”) and in the introduction of section 3 (page 6, “To abstract away the complexity associated with sequential decision making problems, we start by considering environments of length one in this section (i.e., the policy only makes a single prediction) and assume that a labeled dataset is given.”).
>
> The expectation disappears because we assume that the reward function is deterministic. This is a reasonable assumption that holds in most cases and allows us to simplify the equation to:
>
> $Q(s,a) \rightarrow r$
>
> We generalize the Q-function to multiple time steps in section 5.
>
> Our general philosophy is to cover special cases whenever the more general formulation adds complexity without offering additional insight. This is communicated in the Limitations section (page 24, “This article focused on understanding the core ideas of RL. As such, we have focused on intuitive explanations and readable equations instead of theoretical rigor and general equations.”).

---

> > ### Comment · Reviewer_FkiF · 2024-02-19
> >
> > > Simplifying the Q-function to $Q(s,a)\rightarrow r$
> >
> > I still maintain it is an over-simplification and potentially misleading. For example, Section 3.1 states *"The key idea of value learning is to directly predict the reward r rather than the action a and to define the policy π implicitly"* --- I disagree that the value should be presented as the reward like this, without any reference to the fact this is a simplification. I understand this section is meant to emphasize the idea of learning rewards, but I think it does so at the expense of conveying the idea that the value is about the expected future rewards. Even though section 3 starts by saying the presentation simplifies to environments of length one, it's not going to be clear to an introductory reader what this means (e.g., that the value function becomes the reward function).
> >
> > I see two potential solutions here:
> > 1. the easy way is to clarify that the value function in Section 3.1 is in general not the reward function, and reference ahead the generalization in section 5.
> > 2. start with the general multi-step value definitions, as there are not many parts of section 3 that rely on the single-timestep part

---

> > > ### Author Response · Authors · 2024-02-29
> > >
> > > > “My comment is that I don't think it's sensible to train a classification network with Q learning, and it seems reasonable why it's not commonly used today. I still think it's distracting and potentially misleading to keep this section and experiment in the paper”
> > >
> > > Thanks for raising this concern which we would like to understand better. Could you specify why you think it is not sensible to train a classification network with Q learning? The provided Cifar example presents a setting where this is sensible in the sense that the same performance is achieved with the same amount of compute.
> > > To be clear, we do not think that Q-learning is an appropriate choice for every classification task because Q-learning does not scale naively to high dimensional action spaces / large number of classes, whereas stochastic policy gradients do (de Wiele et al. 2020). Instead, we use this example to highlight the connection between reinforcement learning and supervised learning.
> > > In the case that the concern is about classification settings like ImageNet which have a large number of classes then we have now added a clarification to the text to discourage usage in these settings.
> > >
> > > Tom Van de Wiele, David Warde-Farley, Andriy Mnih and Volodymyr Mnih. Q-Learning in enormous action spaces via amortized approximate maximization. arXiv:2001.08116 (2020).
> > >
> > > > To me, the table in S3.1.3 showing that Q-learning improves upon standard supervised classification still strongly claims/implies that this is a research direction worth investigating, or a connection worth learning
> > >
> > > We think i.i.d. classification is close to solved and not a research direction that needs more research. We chose the particular example of accuracy and CIFAR only to demonstrate maximization of a simple non-differentiable metric because it is widely used in teaching and most readers will be familiar with the metric and dataset.
> > > Furthermore, we believe it is important to show the actual quantitative results when presenting an example.
> > > We added a concrete interpretation of these results to the text to make sure that the reader does not interpret these results in the wrong way (e.g. that these results do not imply that Q-learning improves upon standard classification in general).
> > >
> > > > … I see two potential solutions here: …
> > >
> > > Thank you for these suggestions. We implemented suggestion 1.

---

> ### Comment · Reviewer_FkiF · 2024-03-01
>
> > Thanks for raising this concern which we would like to understand better. Could you specify why you think it is not sensible to train a classification network with Q learning?
>
> I read through the Q learning/image classification example again and realize that I had a minor misunderstanding earlier. I understand now that by Q learning in this setting, you simply mean doing MSE regression onto the one-hot label. **To me, this is not Q learning as it does not estimate the expected return of a dynamical system under some policy. I still disagree this is a meaningful connection between RL and supervised learning.**
>
> Perhaps it could be called reward learning if you want to take a bandit perspective on image classification, but even then, I do not see the advantage of making this connection. This is because in most bandit settings, evaluating the arms/actions are difficult and stochastic whereas the image classification reward is deterministic and easy to evaluate for all of the actions.
>
> > Tom Van de Wiele, David Warde-Farley, Andriy Mnih and Volodymyr Mnih. Q-Learning in enormous action spaces via amortized approximate maximization. arXiv:2001.08116 (2020).
>
> I do not think this is an appropriate reference for why MSE regression for image classification might not work in higher dimensions. They are addressing issues arising in Q-learning settings where the value targets are difficult to evaluate in mixed continuous-discrete action spaces. **These issues do not arise when doing classification with MSE onto the one-hot labels.**
> This is one reason why I think the connection is misleading --- the issues papers like that one are pointing out and fixing related to Q-learning/RL often arise due to 1. the sequential nature, 2. inability to evaluate the rewards for all of the actions, or 3. policy learning changing the value targets. All of these issues disappear when doing image classification with MSE.

---

> > ### Author Response · Authors · 2024-03-02
> >
> > > Perhaps it could be called reward learning if you want to take a bandit perspective on image classification, but even then, I do not see the advantage of making this connection. This is because in most bandit settings, evaluating the arms/actions are difficult and stochastic .whereas the image classification reward is deterministic and easy to evaluate for all of the actions
> >
> > The setting in Section 3 can be described as an offline contextual bandit setting, although we have avoided the word bandit since as you mentioned it is typically associated with stochastic rewards and online learning.
> >
> >
> >
> > > To me, this is not Q learning as it does not estimate the expected return of a dynamical system under some policy. I still disagree this is a meaningful connection between RL and supervised learning.
> >
> > As discussed before, the expected return simplifies to the reward in 1-step offline RL settings. We have clarified already that we are considering a special case of Q-learning in section 3. The data collection policy in this setting is a policy that uniformly tries all actions on all images. We don’t exactly see why the idea of Q-learning would require the system to be dynamic. The idea of the Q-function, to extend a value function by condition on the action, which makes the first observed reward independent of the data collection policy and enables the use of the value function to extract the optimal policy, is still present. Additionally, in 1-step environments, the Q-function target becomes independent of the data collection policy by conditioning on the first action (as there are no future actions). We think this is one of the reasons why 1-step RL is significantly easier than standard RL and enables optimization of more complex architectures like ResNet-50. We discuss the numerous additional challenges that the sequential setting imposes in section 5.
> >
> > One reason that this connection is interesting, is that the reader can understand that non-differentiable objectives can be optimized in supervised learning by learning the reward function. The second reason, that this framing is helpful for the reader, is that the way multi-step Q-Learning deals with the problem, that the expected return is not differentiable w.r.t. the policy parameters, is the same as in the 1-step setting (by predicting values instead of actions).
> >
> >
> > > I do not think this is an appropriate reference for why MSE regression for image classification might not work in higher dimensions. They are addressing issues arising in Q-learning settings where the value targets are difficult to evaluate in mixed continuous-discrete action spaces. These issues do not arise when doing classification with MSE onto the one-hot labels. This is one reason why I think the connection is misleading --- the issues papers like that one are pointing out and fixing related to Q-learning/RL often arise due to 1. the sequential nature, 2. inability to evaluate the rewards for all of the actions, or 3. policy learning changing the value targets. All of these issues disappear when doing image classification with MSE.
> >
> > We agree that Q-learning not scaling to high dimensional action spaces in sequential problems doesn’t necessarily imply that it doesn’t scale in 1-step problems, since they are easier.
> >
> > Since the citation is not ideal, we can remove it or reword the text.
> >
> > As a side note, we think that Q-learning also does not scale to high dimensional actions in 1-step settings (at least for classification). We have done some preliminary experiments and Q-learning doesn’t naively scale to ImageNet (learns extremely slow). The technical problem could be different or a subset of the problems observed in sequential settings. The impact of the best action on the average loss decreases with increasing action dimension. In the ImageNet setting, 999 of the actions have reward 0, so predicting 1000x 0 (or uniform distribution) is a shortcut with a very low loss that the network seems to have trouble escaping.

---

> ### Comment · Reviewer_FkiF · 2024-03-02
>
> > We don’t exactly see why the idea of Q-learning would require the system to be dynamic.
>
> I agree, that reward learning and Q learning are the same in the single-step setting. I believe the connection is worth pointing out, but my preference is still to use the term "reward learning" when the model is only predicting rewards and "Q learning" otherwise.
>
> I'm not aware of any other papers that refer to "reward learning" as "Q learning". My understanding is that the distinction between reward and Q learning is usually easy to make. Do you (or the other reviewers) know of any references?
>
> > The data collection policy in this setting is a policy that uniformly tries all actions on all images.
>
> On this, you could also point out that the "replay buffer"/"offline dataset" contains every possible state-action-reward observation under every possible policy. This makes it a very non-standard RL setting. I am also not aware of any other MDP where this is the case. Do you (or the other reviewers) know of any?
>
> > One reason that this connection is interesting, is that the reader can understand that non-differentiable objectives can be optimized in supervised learning by learning the reward function.
>
> As reviewer fykj has also pointed out, supervised learning methods are able to overcome non-differentiability issues without the RL perspective. The setting you're referring to is that the 0-1 classification loss (with uninformative derivatives) can be optimized by instead optimizing the log-likelihood.
>
> Perhaps your main point is that this is a useful introduction to MDPs because it's an MDP perspective on a supervised learning problem people are already familiar with, and the value/policy learning distinction can be clearly shown on it. I still don't think it's a great MDP to teach people the value/policy learning distinction with as it hides the complexities of almost every other MDP arising from the 1. policy, 2. sequential nature, and 3. stochasticity.
>
> > As a side note, we think that Q-learning also does not scale to high dimensional actions in 1-step settings (at least for classification). We have done some preliminary experiments and Q-learning doesn’t naively scale to ImageNet (learns extremely slow). The technical problem could be different or a subset of the problems observed in sequential settings. The impact of the best action on the average loss decreases with increasing action dimension. In the ImageNet setting, 999 of the actions have reward 0, so predicting 1000x 0 (or uniform distribution) is a shortcut with a very low loss that the network seems to have trouble escaping.
>
> As the classes grow, the rewards become sparser. It could probably be fixed by adding back the softmax at the end of the network. From the reward learning perspective this is valid domain knowledge to include, i.e., we know across the entire discrete action space that only one of them is going to have a non-zero reward with a value of 1.

---

> ### Author Response · Authors · 2024-03-03
>
> > I agree, that reward learning and Q learning are the same in the single-step setting. I believe the connection is worth pointing out, but my preference is still to use the term "reward learning" when the model is only predicting rewards and "Q learning" otherwise.
> > I'm not aware of any other papers that refer to "reward learning" as "Q learning". My understanding is that the distinction between reward and Q learning is usually easy to make. Do you (or the other reviewers) know of any references?
>
> We are not aware of other works that point out this connection, which is why we write:
> “While it is well known that classification models can be trained with MSE (Hastie et al., 2009), here we show that this can be viewed as offline Q-learning and leads to policies that maximize accuracy.” (Section 3.1.2).
>
> > On this, you could also point out that the "replay buffer"/"offline dataset" contains every possible state-action-reward observation under every possible policy. This makes it a very non-standard RL setting. I am also not aware of any other MDP where this is the case. Do you (or the other reviewers) know of any?
>
> To be precise, the offline training dataset does not contain every possible observation.
> The training dataset is just a sample of all possible CIFAR like images.
> We use the standard train / validation split and report validation accuracies (training accuracies were also similar for both methods).
> The dataset does contain a reward for every action, which we think makes this setting more sample efficient than usual (in the sense that fewer images are needed for convergence).
>
> > As reviewer fykj has also pointed out, supervised learning methods are able to overcome non-differentiability issues without the RL perspective. The setting you're referring to is that the 0-1 classification loss (with uninformative derivatives) can be optimized by instead optimizing the log-likelihood.
>
> From our understanding, the comment by reviewer fykj was referring to black box optimization methods like evolutionary methods and concerned about the framing in the introduction. Reviewer fykj also highlighted the connections to supervised learning made in section 3 as a particular strength of the draft.
> We discuss the log-likelihood alternative in section 3.2.
> One potential advantage, that the Q-learning (or Q-learning + deterministic policy gradient) approach has, is that it can optimize deterministic policies.
>
> > As the classes grow, the rewards become sparser. It could probably be fixed by adding back the softmax at the end of the network. From the reward learning perspective this is valid domain knowledge to include, i.e., we know across the entire discrete action space that only one of them is going to have a non-zero reward with a value of 1.
>
> Just to be clear, we did not change the architecture, so the experiments already include the softmax. The uniform distribution that networks are typically initialized with have a very good loss if the action dimension becomes large. There is probably some way to address this challenge, which is why we referred to the naive approach.
> As a side note, we also ran some preliminary experiments without the softmax on CIFAR. While that also works, we sometimes observe training instabilities (loss spikes), that the optimization recovers from, but that can lead to a few percent lower final accuracy.

---

### Review · Reviewer_fykj · 2024-02-15

**Summary Of Contributions:**

This manuscript begins by introducing the core concepts of reinforcement learning, aimed at an audience already familiar with supervised learning. It draws several insightful connections between RL and supervised learning. After this introduction, the manuscript enumerates several popular RL methods while pointing out several more low-level details relevant to implementing RL algorithms, including the differences between on-policy and off-policy methods.

**Audience:**

Yes

**Broader Impact Concerns:**

There are no broader impact concerns directly associated with this work, beyond the downstream effects of a readership that becomes potentially more knowledgable about RL.

**Claims And Evidence:**

Yes

**Requested Changes:**

### High-level framing
- Overall, I feel the paper becomes weaker starting around Section 4, where the original motivation of introducing RL in a simple, digestible framing for those already familiar with supervised learning, becomes lost. Instead, the bulk of this 24-page manuscript is actually centered on enumerating a list of RL algorithms while holding the details at arms length. As the original premise of introducing RL to a supervised audience is a fantastic idea (which is why the introductory sections are much stronger), I would strongly suggest greatly parring down these latter sections. One approach to doing so could be to focus on the core challenges of online data collection and credit assignment in a sequential decision-making setting (which are the unique challenges that RL addresses in a principled way), rather than algorithm by algorithm. The current focus on enumerating algorithms rather than focusing on these core challenges places too much emphasis on the implementation details, which seem besides the point of the higher-level motivation of providing a simple, self-contained introduction to RL. Focusing on the core ideas behind the _problem setting_ rather than specific algorithms then opens more opportunities to compare and contrast to the supervised learning setting (a better fit for the intended audience). **I feel the paper should be edited to become more focused in this way before it is published in TMLR, as it would have greater impact in such a form.**

### Comments on the introductory framing:
- I disagree that the key advantage of RL over supervised learning is that RL allows for optimizing non-differentiable objectives. While this is an advantage RL has over supervised learning, it is not a unique one to RL, e.g. blackbox optimization methods in general can address such objectives, including those from the evolutionary computing literature. I believe a more precise framing of the unique aspects of RL should emphasize how RL methods seek to solve the problem of credit assignment, which is an orthogonal problem to simply maximizing an objective. Secondly, and perhaps more importantly, RL methods generate their own data. These two aspects of the RL problem setting actually do not necessarily always occur together, e.g. offline RL (where only credit assignment is the focus) vs. online RL.

- Thus, I believe the introduction could better emphasize more precisely why someone with expertise in supervised learning should invest the time to learn RL, especially when there are supervised alternatives to the classic RLHF pipeline for fine-tuning LLMs (likely one of the biggest motivators for an SL expert to care about RL).

- The discussion in [1] articulates a very similar set of connections between RL and supervised learning, as well as pointing out the connection between offline RL and supervised learning more directly. A concrete example in that discussion also centers on how a supervised learning problem can be recast as an MDP and how RL can then be used to find the optimal solution. It would thus be apt to cite this work, e.g. in the introduction.

- Given the above comments, I think the manuscript would highly benefit from more explicitly structuring the discussion around exploitation and exploration separately. Notably missing in the current exploration discussion is any mention of autocurricula [3,4,5,6] (though self-play, which is a form of this idea, is briefly described, though in passing and somewhat inaccurately).


### Technical concerns
The presentation of certain technical concepts gives me pause:
- In a manuscript that aims to introduce RL, it is surprising that there is no mention of the concept of a Markov decision process.

- In 3.1.2, the reader is left with the impression that Deterministic Policy Gradients (DPG) is the only RL method that can be used for continuous action spaces, when this is not true. Other policy gradients algorithms based on differing formulations, such as PPO, are commonly used for this problem setting.

- Similarly, k-step returns and the discount factor are introduced in either only sections pertaining to off-policy or on-policy algorithms, giving the impression that these concepts are only relevant in those cases respectively, when in reality, these are central ideas that apply generally to the RL problem setting, regardless of this algorithmic distinction.

- The writing is also unclear in many parts. For example, in 3.1.3, "Using the tools of RL in combination with a static dataset is called Offline RL," is vague and likely not meaningful to a reader not already familiar with the problem setting of offline RL. Exactly what constitutes "tools of RL" here is unclear. Is it the MDP formalism? The credit assignment algorithm? This point is actually nuanced, as methods like behavior cloning and Decision Transformer do not directly apply credit assignment algorithms from RL in this setting, while methods like CQL do. At the same time, this is a missed opportunity to draw a useful connection between supervised learning and RL.

- In Section 4.1 there is a somewhat non-sequitur comparison between i.i.d. data sampling and OOD data. These two concepts are orthogonal: You can have OOD data that is also non-i.i.d., e.g. behavioral trajectories sampled from environments different from those seen during training or hyperparameter tuning.

- 4.2 has several statements that are incorrect:
    - Here exploration is defined purely in terms of trying different actions, when exploration is more commonly framed around exploring novel states of the underlying MDP.
    - It is also not clear why the $\epsilon$ parameter is likened to the learning rate.
    - The connection seems to be around how both parameters are annealed during training, but that connection is rather superficial. Moreover, in this section, "entropy bonus" is mentioned but never defined explicitly.
    - This section also claims that performance at training does not matter. This is not true, especially as on many tasks training performance correlates directly with test performance and even extent of exploration.
    - It is stated that a random policy will always lose, when this is incorrect. Given any opponent policy, there is always a non-zero probability for a random policy to sample a trajectory that would be sampled by the best or improving response.

- When discounting is introduced in Section 5, more emphasis should be given to its impact on the variance of the return estimator.


### References
[1] Jiang, Minqi, Tim Rocktäschel, and Edward Grefenstette. "General intelligence requires rethinking exploration." Royal Society Open Science 10.6 (2023): 230539.

[2] Achiam, Joshua. "Spinning up in deep reinforcement learning." (2018).

[3] Leibo, Joel Z., et al. "Autocurricula and the emergence of innovation from social interaction: A manifesto for multi-agent intelligence research." arXiv preprint arXiv:1903.00742 (2019).

[4] Portelas, Rémy, et al. "Automatic curriculum learning for deep rl: A short survey." arXiv preprint arXiv:2003.04664 (2020).

[5] Jiang, Minqi, Edward Grefenstette, and Tim Rocktäschel. "Prioritized level replay." International Conference on Machine Learning. PMLR, 2021.

[6] Dennis, Michael, et al. "Emergent complexity and zero-shot transfer via unsupervised environment design." Advances in neural information processing systems 33 (2020): 13049-13061.

**Strengths And Weaknesses:**

### Strengths
At a high level, the approach of motivating and introducing reinforcement learning fundamentals by drawing direct connections to supervised learning is a fantastic and timely pedogogical framing. The RL problem setting (and associated methods) are seeing a new surge of interest precisely because of their applicability to fine-tuning LLMs, a field full of experts in supervised learning who may have less experience with RL.

### Weaknesses
While the introductory framing presents several insightful connections between RL and supervised learning, this framing dissolves around half-way through Section 4, at which point the manuscript becomes less structured. From this point onwards, the content focuses on describing several subproblems of RL in limited detail, centered around a scattered enumeration of modern deep RL algorithms with only a loose structure, e.g. off-policy vs on-policy. However, many general ideas to RL, such as k-step and discounted returns, are described as specific to one of these subclasses of RL algorithms. This presentation thus comes across as somewhat confusing to a reader familiar with RL and provides an incorrect framing of this information to readers new to the concepts.

Importantly, the fantastic framing of RL for a supervised learning audience disappears entirely at some point in Section 4, making the rest of the paper an enumeration of different RL algorithms, of which excellent and more detailed guides already exist, e.g. Spinning Up RL [2].

---

> ### Author Response · Authors · 2024-02-29
>
> Thank you very much for your detailed feedback and suggestions.
> We have updated the manuscript to address the raised concerns and wanted to clarify some points.
>
> > While the introductory framing presents several insightful connections between RL and supervised learning, this framing dissolves around half-way through Section 4, at which point the manuscript becomes less structured. From this point onwards, the content focuses on describing several subproblems of RL in limited detail, centered around a scattered enumeration of modern deep RL algorithms with only a loose structure, e.g. off-policy vs on-policy.
>
> We are unsure why section 5 and 6 are perceived as unstructured. These sections cover the idea of Q-learning and policy gradients in the sequential decision making settings, respectively. Both sections start with the most basic algorithms, i.e., Q-learning and REINFORCE. We think it is important to cover these algorithms specifically because most modern RL methods can be viewed as extensions to these two fundamental ideas.
>
> Next, both sections cover the limitations of the basic algorithms and popular solutions to these limitations. These solutions are not specific to particular algorithms and are often employed in many modern Deep RL methods.
>
> Lastly, both sections end with one particularly popular modern RL algorithm (section 5.3 and 6.3).
>
> The particular 2 algorithms we present (SAC and PPO) are indeed exchangeable to some extent (e.g. we could also present TD3 and A2C). These subsections are aimed at readers who want to apply RL to their particular problem. We think it is important to cover a sota method for the tutorial to be self-contained and complete, even if other excellent descriptions of these algorithms exist. We added a note, that these are particular examples, and added some pointers to other algorithms.
>
> Reviewer ceMJ also mentions these subsections as a significant strength of the tutorial.
>
> An additional purpose of these subsections is to make a recommendation to the reader what algorithm we think is a reasonable choice to start with on a new problem. This question can otherwise be overwhelming given the large range of existing RL methods.
>
> > However, many general ideas to RL, such as k-step and discounted returns, are described as specific to one of these subclasses of RL algorithms.
>
> We do mention that some of the discussed ideas are more general “While we discuss these ideas in the context of policy gradients, some of them can also be applied to value learning methods” (section 6, page 17) but we agree that this should be made more precise.
> In particular we added explicitly that discount factors and reward shaping can be used with basically any RL algorithm and k-step returns / eligibility traces with any RL algorithm that uses a Value/Q-function. Thanks for the suggestion.
>
> > …I believe a more precise framing of the unique aspects of RL should emphasize how RL methods seek to solve the problem of credit assignment, which is an orthogonal problem to simply maximizing an objective. …
>
> We respectfully disagree with the point that RL methods generally seek to solve the problem of credit assignment. We think that RL methods aim to find the optimal policy for a given environment and metric / MDP. Finding this optimal policy necessitates both data collection and optimization. Solving the credit assignment problem while useful is optional. An example of an RL method that does not solve the credit assignment problem would be the vanilla REINFORCE algorithm. The optimal policy only requires finding the optimal action, but does not necessarily need to know how much better that action is or how much it will contribute to the return.
>
> We see the utility of solving the credit assignment problem in that it will make optimization significantly easier. We added a brief mention of this in section 6. For example, finding the true advantage function can significantly reduce the variance of policy gradient methods. Similarly, finding the Q-function of the optimal policy simplifies optimization to maximizing the Q-function.
> As such we see solving the problem of credit assignment to be complementary not orthogonal to reward maximization.
>
> Due to that reasoning, we chose to present credit assignment techniques as solutions to optimization problems in section 5 and 6.

---

> > ### Author Response · Authors · 2024-02-29
> >
> > > I disagree that the key advantage of RL over supervised learning is that RL allows for optimizing non-differentiable objectives. While this is an advantage RL has over supervised learning, it is not a unique one to RL, e.g. blackbox optimization methods in general can address such objectives, including those from the evolutionary computing literature. I believe a more precise framing of the unique aspects of RL should emphasize how RL methods seek to solve the problem of credit assignment, which is an orthogonal problem to simply maximizing an objective.
> >
> > Thank you for the comment. We have now made the framing weaker to make clear that it is one of the key advantages of RL.
> > Note that our framing does not imply that there are no other methods that can optimize non-differentiable rewards (though in practice black box optimization and evolutionary methods tend to be too inefficient to optimize Deep Neural Networks). We added a clarification to the introduction.
> >
> >
> > > Secondly, and perhaps more importantly, RL methods generate their own data. These two aspects of the RL problem setting actually do not necessarily always occur together, e.g. offline RL (where only credit assignment is the focus) vs. online RL.
> >
> > Thanks for the comment. We have now added a sentence to highlight automatic data collection as a second advantage of RL methods.
> >
> > > … It would thus be apt to cite this work, e.g. in the introduction.
> >
> > Thanks, we added the citation.
> >
> > > Given the above comments, I think the manuscript would highly benefit from more explicitly structuring the discussion around exploitation and exploration separately. Notably missing in the current exploration discussion is any mention of autocurricula [3,4,5,6] (though self-play, which is a form of this idea, is briefly described, though in passing and somewhat inaccurately).
> >
> > Currently the discussion on exploration and exploitation is divided into paragraphs. It is a bit unclear to us how we should improve the manuscript by separating the topics further. We would appreciate a specific suggestion that will help us improve the manuscript.
> >
> > Thank you for the suggestion on curricula. We have now added a paragraph discussing curriculum learning. We chose to focus the discussion on the more traditional “engineered” curriculum approach since this idea is already at the level of maturity where it is used in successful real world applications of RL (Lee et al. 2020, Miki et al. 2022). We restricted the discussion of automatic curriculum learning to some pointers to the literature for the interested reader, since it seems that the community hasn’t converged on a standard method yet (that withstood the test of time).
> >
> > Joonho Lee, Jemin Hwangbo, Lorenz Wellhausen, Vladlen Koltun, and Marco Hutter. Learning quadrupedal locomotion over challenging terrain. Science Robotics, 2020.
> >
> > Takahiro Miki, Joonho Lee, Jemin Hwangbo, Lorenz Wellhausen, Vladlen Koltun, and Marco Hutter. Learning robust perceptive locomotion for quadrupedal robots in the wild. Science Robotics, 2022.
> >
> > > In a manuscript that aims to introduce RL, it is surprising that there is no mention of the concept of a Markov decision process.
> >
> > As stated in the introduction, this is intentional since the MDP is not necessary to understand the contents of the paper while adding additional notation.
> >
> > We think the MDP is a helpful formalism for theoretical research but to some extent overused in experimental works, where neither the initial states, the transition kernel nor the set of all states are known (and it is often questionable whether states strictly have the Markov property, stacked frames on Atari being an example).
> >
> > Since theoretical researchers are not the target audience for this draft, we haven’t included it so far.
> >
> > That being said, we agree that there is an argument to be made that introducing the MDP, while not necessary for this paper, will help the reader understand other papers, since the concept is so broadly used.
> > We have hence added a paragraph in the notation section mentioning the MDP.

---

> > > ### Author Response · Authors · 2024-02-29
> > >
> > > > In 3.1.2, the reader is left with the impression that Deterministic Policy Gradients (DPG) is the only RL method that can be used for continuous action spaces, when this is not true. Other policy gradients algorithms based on differing formulations, such as PPO, are commonly used for this problem setting.
> > >
> > > Note that we state that DPG is a “A common solution to this problem…” (Page 8) not the only one.
> > > We have now made it more clear that this specifically refers to Q-learning based methods.
> > > For policy gradient methods like PPO, we mention (at the end of section 3.2) that they can naively be used for continuous action spaces by parameterizing a continuous probability distribution.
> > >
> > > > Similarly, k-step returns and the discount factor are introduced in either only sections pertaining to off-policy or on-policy algorithms, giving the impression that these concepts are only relevant in those cases respectively, when in reality, these are central ideas that apply generally to the RL problem setting, regardless of this algorithmic distinction.
> > >
> > > Thanks for the comment. We have now made it more clear in the text that these specific methods can be applied more broadly.
> > >
> > > > The writing is also unclear in many parts. For example, in 3.1.3, "Using the tools of RL in combination with a static dataset is called Offline RL," is vague and likely not meaningful to a reader not already familiar with the problem setting of offline RL. Exactly what constitutes "tools of RL" here is unclear. Is it the MDP formalism? The credit assignment algorithm? This point is actually nuanced, as methods like behavior cloning and Decision Transformer do not directly apply credit assignment algorithms from RL in this setting, while methods like CQL do. At the same time, this is a missed opportunity to draw a useful connection between supervised learning and RL.
> > >
> > > Thanks, we have now made this more clear by  referring to the optimization techniques of RL. As argued above, we consider credit assignment techniques as a part of optimization.
> > >
> > > > In Section 4.1 there is a somewhat non-sequitur comparison between i.i.d. data sampling and OOD data. These two concepts are orthogonal: You can have OOD data that is also non-i.i.d., e.g. behavioral trajectories sampled from environments different from those seen during training or hyperparameter tuning.
> > >
> > > Thank you for the suggestion, we have changed the name to Non-IID, to make it clear that also the independence assumption might be violated.
> > >
> > > > Here exploration is defined purely in terms of trying different actions, when exploration is more commonly framed around exploring novel states of the underlying MDP.
> > >
> > > It is unclear to us why this is perceived that way. Note that we explicitly state that the purpose of exploration is to find novel states: “However, to discover new states, other actions must sometimes be chosen. This is called exploration.”(sec. 4.2, page 11). In the case that this feels insufficient, a concrete suggestion for rephrasing would be appreciated.
> > >
> > > > Moreover, in this section, "entropy bonus" is mentioned but never defined explicitly.
> > >
> > > We have now clarified that an objective to increase the entropy of the policy was meant.
> > >
> > > > This section also claims that performance at training does not matter. This is not true, especially as on many tasks training performance correlates directly with test performance and even extent of exploration.
> > >
> > > Thanks. We have now clarified that we were referring to the actual rewards obtained during the training phase, not the quality of the policy. These can be decoupled, for example in the Go-Explore methods (Ecoffet et al. 2021). Additionally, we are referring to a setting where e.g. a practitioner wants to train and deploy an agent. The total rewards obtained during the training phase usually don’t matter for the designer of the algorithm, whereas rewards obtained during deployment do. (I.e. it only matters how good the final policy is).
> > >
> > > Adrien Ecoffet, Joost Huizinga, Joel Lehman, Kenneth O. Stanley, and Jeff Clune. First return, then explore. Nature, 2021
> > >
> > > > It is stated that a random policy will always lose, when this is incorrect. Given any opponent policy, there is always a non-zero probability for a random policy to sample a trajectory that would be sampled by the best or improving response.
> > >
> > > We have changed the wording to “extremely unlikely to win” and clarified that sampling improved actions will not help optimization due to the sparse reward in this setting.
> > >
> > > > When discounting is introduced in Section 5, more emphasis should be given to its impact on the variance of the return estimator.
> > >
> > > Thank you for the suggestion. We added additional emphasis.

---

### Review · Reviewer_ceMJ · 2024-02-19

**Summary Of Contributions:**

The submission is a comprehensive tutorial designed to bridge the knowledge gap for data scientists familiar with Supervised Learning, guiding them towards a robust understanding of Deep Reinforcement Learning (DRL). This paper effectively demystifies the complexities of Reinforcement Learning (RL) by breaking down its problems into manageable segments, particularly emphasizing One Step and Episodic learning approaches in Section 3. It introduces readers to essential data collection techniques and delineates the distinctions between Off Policy and On Policy RL algorithms, providing a foundational understanding necessary for delving into more advanced topics.

The tutorial meticulously addresses the inherent challenges associated with Off Policy learning, presenting various methodologies in Section 5.2 to navigate these obstacles effectively. Similarly, it delves into the On Policy algorithm REINFORCE, discussing the issues of variance and sample inefficiency that often plague this approach. Section 6.2 is devoted to outlining strategies to mitigate these challenges, ensuring a comprehensive understanding of the algorithm's nuances.

One of the paper's significant strengths lies in its exposition of state-of-the-art RL algorithms, such as Soft Actor Critic and Proximal Policy Optimization. By providing insights into these advanced methodologies, the paper not only educates its audience on the current frontiers of RL research but also equips them with the knowledge to apply these cutting-edge techniques in practical scenarios.

In summary, this tutorial paper stands out as an invaluable resource for data scientists seeking to transition from Supervised Learning paradigms to the more dynamic and complex domain of Deep Reinforcement Learning.

**Audience:**

Yes

**Broader Impact Concerns:**

At this point of time, I cannot think of any concerns on the ethical implications of the work that would require adding a Broader Impact Statement

**Claims And Evidence:**

Yes

**Requested Changes:**

Example Placement: Consider relocating the example in Section 3.1.3 to a more suitable position, such as after the discussion on Discrete Action Space Reinforcement Learning in Section 3.1.1. This adjustment can enhance the flow and clarity of the content.

Bellman Equations Primer: Adding a primer on Bellman Equations, either in Section 2 within the Notations discussion or in an Appendix, would be beneficial for readers new to the field of Reinforcement Learning. This addition can provide essential foundational knowledge for better understanding.

Clarification on Soft Actor Critic Equations: In Section 5.3, clarity is needed on how equation 21 is derived from equation 20 for Soft Actor Critic. Additionally, discussing the Entropy function could further elucidate this aspect of the algorithm.

Policy Gradient Theorem Explanation: To enhance the explanation of the Policy Gradient theorem and its application in calculating update gradients, referencing tutorials by Professor Sergey Levine, such as the provided links, could offer valuable insights.
 (https://www.youtube.com/watch?v=GKoKNYaBvM0&list=PL_iWQOsE6TfXxKgI1GgyV1B_Xa0DxE5eH&index=22, https://rail.eecs.berkeley.edu/deeprlcourse/deeprlcourse/static/homeworks/hw2.pdf)

**Strengths And Weaknesses:**

Strengths:
Comprehensive Overview: The submission provides a thorough and detailed overview of Deep Reinforcement Learning (RL), catering to individuals familiar with Supervised Learning . This comprehensive coverage ensures that readers gain a solid understanding of the subject matter.

Structured Approach: The tutorial paper systematically breaks down the complexities of RL into manageable components, starting with One Step Episodic and progressing to Off Policy and On Policy RL Algorithms. This structured approach aids in enhancing the audience's comprehension .

Inclusion of Advanced Algorithms: By introducing cutting-edge RL algorithms like Soft Actor Critic and Proximal Policy Optimization, the submission exposes readers to state-of-the-art techniques in the field. This inclusion broadens the readers' knowledge and provides insights into the latest developments in RL methodologies .

Weaknesses:
Example Placement: Consider relocating the example in Section 3.1.3 to a more suitable position, such as after the discussion on Discrete Action Space Reinforcement Learning in Section 3.1.1. This adjustment can enhance the flow and clarity of the content.

Bellman Equations Primer: Adding a primer on Bellman Equations, either in Section 2 within the Notations discussion or in an Appendix, would be beneficial for readers new to the field of Reinforcement Learning. This addition can provide essential foundational knowledge for better understanding.

Clarification on Soft Actor Critic Equations: In Section 5.3, clarity is needed on how equation 21 is derived from equation 20 for Soft Actor Critic. Additionally, discussing the Entropy function could further elucidate this aspect of the algorithm.

Policy Gradient Theorem Explanation: To enhance the explanation of the Policy Gradient theorem and its application in calculating update gradients, referencing tutorials by Professor Sergey Levine, such as the provided links, could offer valuable insights.

In summary, while the submission excels in providing a comprehensive overview and structured approach to Deep Reinforcement Learning, addressing the weaknesses related to example placement, Bellman Equations primer, clarification on Soft Actor Critic equations, and Policy Gradient Theorem explanation can further enhance the clarity and depth of the content.

---

> ### Author Response · Authors · 2024-02-29
>
> > Example Placement: Consider relocating the example in Section 3.1.3 to a more suitable position, such as after the discussion on Discrete Action Space Reinforcement Learning in Section 3.1.1. This adjustment can enhance the flow and clarity of the content.
>
> Thank you for the suggestion, we moved section 3.1.3 after section 3.1.1.
>
> > Bellman Equations Primer: Adding a primer on Bellman Equations, either in Section 2 within the Notations discussion or in an Appendix, would be beneficial for readers new to the field of Reinforcement Learning. This addition can provide essential foundational knowledge for better understanding.
>
> It is not exactly clear to us what is envisioned here. We do introduce a Bellman Equation for the Q-function with deterministic rewards and environments in Equation 12, Page 14.
> Is the reviewer asking for another form of the Bellman Equation or more explanation?
>
> > Clarification on Soft Actor Critic Equations: In Section 5.3, clarity is needed on how equation 21 is derived from equation 20 for Soft Actor Critic. Additionally, discussing the Entropy function could further elucidate this aspect of the algorithm.
>
> We added the definition of the entropy function. The exact way how the entropy is incorporated into the Q-function (here for the next time step not the current one) is indeed relatively arbitrary in Soft-Actor Critic (Achiam 2018) so there is no concrete explanation (in our opinion).
> Haarnoja et al. (2018) just define the entropy augmented reward to include the entropy at the next time step in Appendix B.
> One could argue that this definition is more computationally efficient than using the entropy of the current time step, since only one forward pass through the policy is required in Equation 21.
>
> Joshua Achiam. Spinning Up in Deep Reinforcement Learning. url: https://spinningup.openai.com, 2018.
>
> Tuomas Haarnoja, Aurick Zhou, Pieter Abbeel, and Sergey Levine. Soft actor-critic: Off-policy maximum entropy deep reinforcement learning with a stochastic actor. In Proc. of the International Conf. on Machine learning (ICML), 2018
>
> > Policy Gradient Theorem Explanation: To enhance the explanation of the Policy Gradient theorem and its application in calculating update gradients, referencing tutorials by Professor Sergey Levine, such as the provided links, could offer valuable insights.
>
> Thanks for the suggestion. We have added  a citation to the lecture.

---

> > ### Comment · Reviewer_ceMJ · 2024-03-02
> > **Clarification on Bellman Equations Primer**
> >
> > Hi
> >
> > Thank you for incorporating the changes
> >
> > Clarifying on following
> >
> > Bellman Equations Primer: Adding a primer on Bellman Equations, either in Section 2 within the Notations discussion or in an Appendix, would be beneficial for readers new to the field of Reinforcement Learning. This addition can provide essential foundational knowledge for better understanding.
> >
> > Suggested Content for the Bellman Equations Primer: Introduction to Policies and Value Functions: Begin with an overview of policies and value functions, crucial components in RL. A compelling example to illustrate this concept can be found in Sutton and Barto's "Reinforcement Learning: An Introduction" (2nd ed.), particularly discussing equations 3.12 and 3.13. This section should explain how policies dictate the behavior of an agent within an environment and how value functions estimate the expected return from states or state-action pairs under a specific policy.
> >
> > Recursive Relation between Value Functions: Delve into the recursive nature of value functions, highlighting their dependency on subsequent states' value functions. Sutton and Barto's discussion around equation 3.14 serves as an excellent example of this concept. This recursive relationship is foundational to understanding how RL algorithms bootstrap their estimates to learn optimal policies over time.
> >
> > Bellman's Optimality Equations: Finally, address the concept of Bellman's Optimality Equations, focusing on how these equations define the relationship between the value of a state and the values of subsequent states for optimal policies. An insightful discussion on this topic is also available in Sutton and Barto's work, specifically in the section on "Optimal Policies and Optimal Value Functions" (Chapter 3.6). This part should elucidate how these equations serve as the cornerstone for many RL algorithms, guiding the search for optimal policies that maximize the expected return.

---

> > > ### Author Response · Authors · 2024-03-03
> > >
> > > Thank you for the clarification. We have added a primer for the Bellman Equations to the Appendix.

---

### Author Response · Authors · 2024-02-29

We thank all reviewers for taking the time to read our manuscript and for their valuable feedback.

The reviewers pointed out positively the pedagogical framing of the tutorial, introducing RL by drawing connections to supervised learning, the structured approach, breaking down the complexities of RL into manageable components and the inclusion of advanced algorithms.

The reviewers requested several changes. We have updated our manuscript to address these concerns. Our changes are highlighted in red color.

Reviewer FkiF pointed out ways in which a reader might misinterpret the text. We added clarifying sentences to the text to prevent this.

Reviewer fykj made comments on the framing of the paper and contributed comments on many low level details.
We have incorporated many of these comments, including the addition of a paragraph on curriculum learning, and discuss some points that were unclear below.

Reviewer ceMJ made some comments on example placement, which we incorporated.

We respond in more detail to each of the raised points below.

---

### Decision · Action_Editor_YWSR · 2024-03-29

**Recommendation:** Reject

**Comment:**

The reviews were a mixed bag. The crux of the recommendations for rejection were that too much of this lecture (mainly the last 75% of the paper, section 4 onwards) is just a traditional introduction to RL, and that what made the first few sections novel and interesting (namely describing RL using SL concepts) was somewhat lost, either indicating that these sections should be substantially rewritten to incorporate this aspect, or dropped. Despite a number of technical and presentation matters being addressed by the authors following the reviews, it looks like a large part of the paper does not add much to the number of excellent surveys of RL methods online and in print. I think that the majority of reviewers would like to see this published if these sections were written more following the design principles that make section 3 so worth reading, and I would recommend that they authors submit a major revision at a later time after conducting this work.

**Audience:**

This would be of interest to a number of ML specialists with experience in supervised methods, and little-to-no experience in RL.

**Claims And Evidence:**

N/A

This paper is a tutorial/set of lecture notes introducing RL to a more SL-oriented readership, casting RL into a conceptual framework those readers will be more familiar with, in an effort to demystify certain aspects of RL.

**Resubmission Of Major Revision:**

The authors may consider submitting a major revision at a later time.